# Measuring Robustness to Natural Distribution Shifts in Image Classification

**Rohan Taori**
UC Berkeley

**Achal Dave**
CMU

**Vaishaal Shankar**
UC Berkeley

**Nicholas Carlini**
Google Brain

**Benjamin Recht**
UC Berkeley

**Ludwig Schmidt**
UC Berkeley

## Abstract

We study how robust current ImageNet models are to distribution shifts arising from natural variations in datasets. Most research on robustness focuses on synthetic image perturbations (noise, simulated weather artifacts, adversarial examples, etc.), which leaves open how robustness on synthetic distribution shift relates to distribution shift arising in real data. Informed by an evaluation of 204 ImageNet models in 213 different test conditions, we find that there is often little to no transfer of robustness from current synthetic to natural distribution shift. Moreover, most current techniques provide no robustness to the natural distribution shifts in our testbed. The main exception is training on larger and more diverse datasets, which in multiple cases increases robustness, but is still far from closing the performance gaps. Our results indicate that distribution shifts arising in real data are currently an open research problem. We provide our testbed and data as a resource for future work at https://modestyachts.github.io/imagenet-testbed/.

## 1 Introduction

Reliable classification under distribution shift is still out of reach for current machine learning [65, 68, 91]. As a result, the research community has proposed a wide range of evaluation protocols that go beyond a single, static test set. Common examples include noise corruptions [33, 38], spatial transformations [28, 29], and adversarial examples [5, 84]. Encouragingly, the past few years have seen substantial progress in robustness to these distribution shifts, e.g., see [13, 28, 34, 55, 57, 66, 93, 96, 105, 114, 115] among many others. However, this progress comes with an important limitation: all of the aforementioned distribution shifts are *synthetic*: the test examples are derived from well-characterized image modifications at the pixel level.

Synthetic distribution shifts are a good starting point for experiments since they are precisely defined and easy to apply to arbitrary images. However, classifiers ultimately must be robust to distribution shifts arising naturally in the real world. These distribution shifts may include subtle changes in scene compositions, object types, lighting conditions, and many others. Importantly, these variations are *not* precisely defined because they have not been created artificially. The hope is that an ideal robust classifier is still robust to such natural distribution shifts.

In this paper, we investigate how robust current machine learning techniques are to distribution shift arising naturally from real image data without synthetic modifications. To this end, we conduct a comprehensive experimental study in the context of ImageNet [18, 70]. ImageNet is a natural starting point since it has been the focus of intense research efforts over the past decade and a large number of pre-trained classification models, some with robustness interventions, are available for this task. The core of our experimental study is a testbed of 204 pre-trained ImageNet models that we evaluate in 213 different settings, covering both the most popular models and distribution shifts. Our testbed consists of $10^9$ model predictions and is 100 times larger than prior work [27, 33, 47, 68]. This allows us to draw several new conclusions about current robustness interventions:

**Robustness measurements should control for accuracy.** Existing work typically argues that an intervention improves robustness by showing that the accuracy on a robustness test set has improved (e.g., see [34, 40, 63, 102, 115]). We find that in many cases, this improved robustness can be explained by the model performing better on the standard, unperturbed test set. For instance, using different model architectures does not substantially improve the robustness of a model beyond what would be expected from having a higher standard accuracy. While training more accurate models is clearly useful, it is important to separate accuracy improvements from robustness improvements when interpreting the results.

**Current synthetic robustness measures do not imply natural robustness.** Prior work often evaluates on synthetic distribution shifts to measure robustness [9, 32, 38]. We find that current robustness measures for synthetic distribution shift are at most weakly predictive for robustness on the natural distribution shifts presently available. While there are good reasons to study synthetic forms of robustness – for instance, adversarial examples are interesting from a security perspective – synthetic distribution shifts alone do not provide a comprehensive measure of robustness at this time. Moreover, as the right plot in Figure 1 exemplifies, current robustness interventions are often (but not always) ineffective on the natural distribution shifts in our testbed.

**Training on more diverse data improves robustness.** Across all of our experiments, the only intervention that improves robustness to multiple natural distribution shifts is training with a more diverse dataset. This overarching trend has not previously been identified and stands out only through our large testbed. Quantifying when and why training with more data helps is an interesting open question: while more data is generally helpful, we find some models that are trained on 100 times more data than the standard ImageNet training set but do not provide any robustness.

The goal of our paper is specifically *not* to introduce a new classification method or image dataset. Instead, our paper is a meta-study of current robustness research to identify overarching trends that span multiple evaluation settings. This is particularly important if the ultimate goal of a research direction is to produce models that function reliably in a wide variety of contexts. Our findings highlight robustness on real data as a clear challenge for future work. Due to the diminishing returns of larger training datasets, addressing this robustness challenge will likely require new algorithmic ideas and more evaluations on natural distribution shifts.

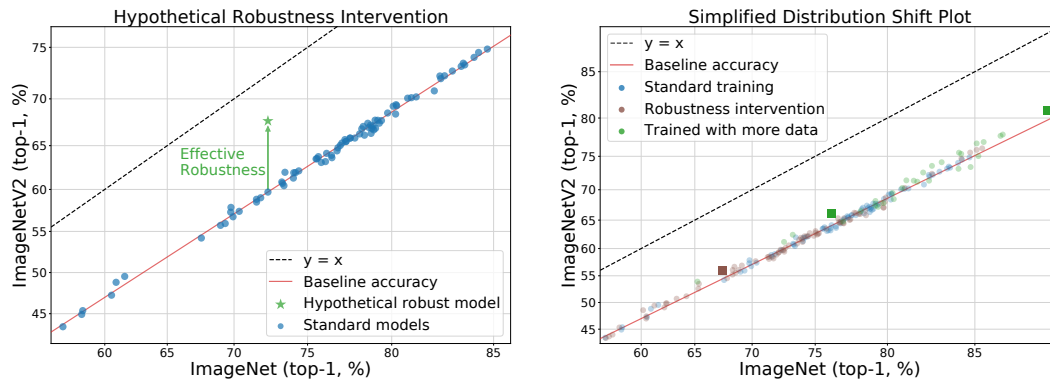

Figure 1: (Left) We plot 78 standard models trained on ImageNet without any robustness interventions, showing both their accuracy on the standard test set (ImageNet, x-axis) and on a test set with distribution shift (ImageNetV2, y-axis). All models lie below the $y = x$ line: their accuracy under this distribution shift is lower than on the standard test set. Nevertheless, improvements in accuracy on the standard test set almost perfectly predict a consistent improvement under distribution shift, as shown by the linear fit (red line). A hypothetical robustness intervention, shown in green, should provide *effective robustness*, i.e., the intervention should improve the accuracy under distribution shift beyond what is predicted by the linear fit.
(Right) We plot most of the 204 models in our testbed, highlighting those with the highest effective robustness using square markers. These models are still far from closing the accuracy gap induced by the distribution shift (ideally a robust model would fall on the $y = x$ line). Figure 2 shows a more detailed version of this plot with error bars for all points.

## 2 Measuring robustness

We first discuss how to measure robustness as a quantity distinct from accuracy. In our experiments, we always have two evaluation settings: the "standard" test set, and the test set with distribution shift. For a model $f$, we denote the two accuracies with $\text{acc}_1(f)$ and $\text{acc}_2(f)$, respectively.

When comparing the robustness of two models $f_a$ and $f_b$, one approach would be to rank the models by their accuracy under distribution shift. However, this approach does not disentangle the robustness of a model from its accuracy on the standard test set. As an example, consider a pair of models with accuracy $\text{acc}_1(f_a) = 0.8$, $\text{acc}_2(f_a) = 0.75$ (i.e., a 5% drop in accuracy from the distribution shift), and $\text{acc}_1(f_b) = 0.9$, $\text{acc}_2(f_b) = 0.76$ (a 14% drop). Model $f_b$ has higher accuracy on the second test set, but overall sees a drop of 14% from the standard to the shifted test set. In contrast, the first model sees only a 5% drop. Hence we would like to refer to the first model as more robust, even though it achieves lower accuracy on the shifted test set.

**Effective robustness.** The core issue in the preceding example is that standard accuracy ($\text{acc}_1$) acts as a confounder. Instead of directly comparing accuracies under distribution shift, we would like to understand if a model $f_b$ offers higher accuracy on the shifted test set *beyond what is expected from having higher accuracy on the original test set*. We call this notion of robustness beyond a baseline *effective robustness*. Graphically, effective robustness corresponds to a model being above the linear trend (red line) given by our testbed of standard models in Figure 1 (left).

To precisely define effective robustness, we introduce $\beta(x)$, the baseline accuracy on the shifted test set for a given accuracy $x$ on the standard test set. On the distribution shifts in our testbed, we instantiate $\beta$ by computing the parameters of a log-linear fit for the models without a robustness intervention (the red line in Figure 1). Empirically, this approach yields a good fit to the data. For other distribution shifts, the baseline accuracy may follow different trends and may also depend on properties beyond the standard accuracy, e.g., model architecture. Appendix J.1 contains detailed information on how to compute $\beta$.

Given the accuracy baseline $\beta$, we define the effective robustness of a model as

$$\rho(f) = \text{acc}_2(f) - \beta(\text{acc}_1(f)) .$$

A model without special robustness properties falls on the linear fit and hence has $\rho(f) = 0$. The main goal of a robustness intervention is to increase $\rho$. Models with large $\rho$ offer robustness beyond what we can currently achieve with standard models.

**Relative robustness.** Effective robustness alone does not imply that a robustness intervention is useful. In particular, a robustness intervention could increase $\rho$ for a model it is applied to, but at the same time *decrease* both $\text{acc}_1$ and $\text{acc}_2$. Such a robustness intervention would offer no benefits. So to complement effective robustness, we also introduce *relative robustness* to directly quantify the effect of an intervention on the accuracy under distribution shift. For a model $f'$ with robustness intervention, derived from a model $f$ without the intervention, the relative robustness is $\tau(f') = \text{acc}_2(f') - \text{acc}_2(f)$. We graphically illustrate this notion of robustness in Appendix C.1.

Overall, a useful robustness intervention should obtain *both* positive effective and relative robustness. As we will see, only few classification models currently achieve this goal, and no models achieve both large effective and relative robustness.

## 3 Experimental setup

We now describe our experimental setup. A model $f$ is first trained on a fixed training set. We then evaluate this model on two test sets: the "standard" test set (denoted $S_1$) and the test set with a distribution shift (denoted $S_2$).

A crucial question in this setup is what accuracy the model $f$ can possibly achieve on the test set with distribution shift. In order to ensure that the accuracy on the two test sets are comparable, we focus on natural distribution shifts where humans have thoroughly reviewed the test sets to include only correctly labeled images [2, 18, 39, 68, 76].[1] This implies that an ideal robust classifier does not have a substantial accuracy gap between the two test sets. Indeed, recent work experimentally

confirms that humans achieve similar classification accuracy on the original ImageNet test set and the ImageNetV2 replication study (one of the distribution shifts in our testbed) [77].

## 3.1 Types of distribution shifts

At a high level, we distinguish between two main types of distribution shift. We use the term *natural* distribution shift for datasets that rely only on unmodified images. In contrast, we refer to distribution shifts as *synthetic* if they involve modifications of existing images specifically to test robustness. To be concrete, we next provide an overview of the distribution shifts in our robustness evaluation, with further details in Appendix F and visual overviews in Appendices A and K.

### 3.1.1 Natural distribution shifts

We evaluate on seven natural distribution shifts that we classify into three categories.

**Consistency shifts.** To evaluate a notion of robustness similar to $\ell_p$-adversarial examples but without synthetic perturbations, we measure robustness to small changes across video frames as introduced by Gu et al. [35] and Shankar et al. [76]. The authors assembled sets of contiguous video frames that appear perceptually similar to humans, but produce inconsistent predictions for classifiers. We define $S_1$ to be the set of "anchor" frames in each video, and evaluate the accuracy under distribution shift by choosing the worst frame from each frame set for a classifier. This is the "pm-k" metric introduced by Shankar et al. [76].

**Dataset shifts.** Next, we consider datasets $S_2$ that are collected in a different manner from $S_1$ but still evaluate a classification task with a compatible set of classes. These distribution shifts test to what extent current robustness interventions help with natural variations between datasets that are hard to model explicitly. We consider four datasets of this variety: (i) ImageNetV2, a reproduction of the ImageNet test set collected by Recht et al. [68]; (ii) ObjectNet, a test set of objects in a variety of scenes with 113 classes that overlap with ImageNet [2]; and, (iii) ImageNetVid-Robust-anchor and YTBB-Robust-anchor [76], which are the datasets constructed from only the anchor frames in the consistency datasets described above. These two datasets contain 30 and 24 super-classes of the ImageNet class hierarchy, respectively. For each of these distribution shifts, we define $S_1$ to be a subset of the ImageNet test set with the same label set as $S_2$ so that the accuracies are comparable.

**Adversarially filtered shifts.** Finally, we consider an adversarially collected dataset, ImageNet-A [39]. Hendrycks et al. [39] assembled the dataset by downloading a large number of labeled images from Flickr, DuckDuckGo, iNaturalist, and other sites, and then selected the subset that was misclassified by a ResNet-50 model. We include ImageNet-A in our testbed to investigate whether the adversarial filtering process leads to qualitatively different results. Since ImageNet-A contains only 200 classes, the standard test set $S_1$ here is again a subset of the ImageNet test set that has the same 200 classes as ImageNet-A.

### 3.1.2 Synthetic distribution shifts

The research community has developed a wide range of synthetic robustness notions for image classification over the past five years. In our study, we consider the following classes of synthetic distribution shifts, which cover the most common types of image perturbations.

**Image corruptions.** We include all corruptions from [38], as well as some corruptions from [33]. These include common examples of image noise (Gaussian, shot noise), various blurs (Gaussian, motion), simulated weather conditions (fog, snow), and "digital" corruptions such as various JPEG compression levels. We refer the reader to Appendix F.2 for a full list of the 38 corruptions.

**Style transfer.** We use a stylized version of the ImageNet test set [34, 44].

**Adversarial examples.** We include untargeted adversarial perturbations bounded in $\ell_\infty$- or $\ell_2$-norm by running projected gradient descent as described in [55]. We use $\varepsilon = \{\frac{0.5}{255}, \frac{2}{255}\}$ for $\ell_\infty$ and $\varepsilon = \{0.1, 0.5\}$ for $\ell_2$ (further details in Appendix F.3).

## 3.2 Classification models

Our model testbed includes 204 ImageNet models covering a variety of different architectures and training methods. The models can be divided into the following three categories (see Appendix G for a full list of all models and their categories).

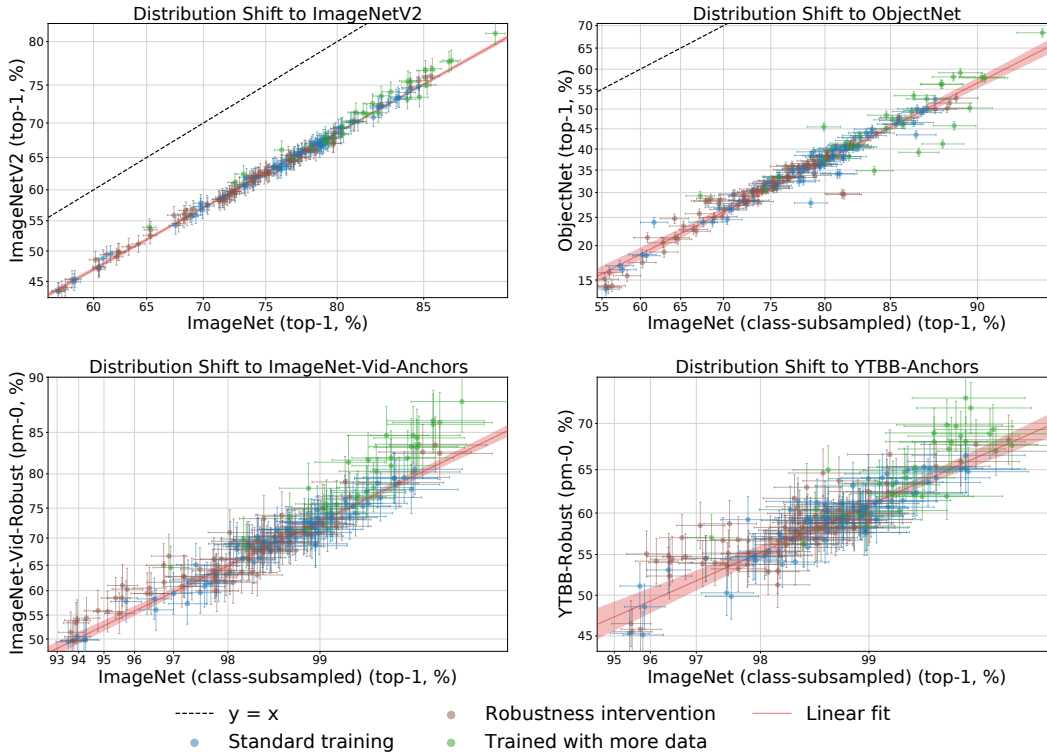

Figure 2: Model accuracies on the four natural dataset shifts: ImageNetV2 (top left), ObjectNet (top right), ImageNet-Vid-Robust-anchor (bottom left), and YTBB-Robust-anchor (bottom right). These plots demonstrate that the standard test accuracy (x-axis) is a reliable predictor for the test accuracy under distribution shift (y-axis), especially for models trained without a robustness intervention. The notable outliers to this trend are some models trained on substantially more data. For ObjectNet, ImageNet-Vid-Robust-anchor, and YTBB-Robust-anchor, we show the accuracy on a subset of the ImageNet classes on the x-axis to match the label space of the target task (y-axis). Each data point corresponds to one model in our testbed and is shown with 99.5% Clopper-Pearson confidence intervals. The axes were adjusted using logit scaling and the linear fit was computed in the scaled space on only the standard models. The red shaded region is a 95% confidence region for the linear fit from 1,000 bootstrap samples.

**Standard models.** We refer to models trained on the ILSVRC 2012 training set without a specific robustness focus as *standard* models. This category includes 78 models with architectures ranging from AlexNet to EfficietNet, e.g., [37, 50, 78, 85, 88].

**Robust models.** This category includes 86 models with an explicit robustness intervention such as adversarially robust models [13, 27, 72, 74, 101], models with special data augmentation [20, 28, 34, 41, 100, 108, 113], and models with architecture modifications [115].

**Models trained on more data.** Finally, our testbed contains 30 models that utilize substantially more training data than the standard ImageNet training set. This subset includes models trained on (i) Facebook's collection of 1 billion Instagram images [56, 104], (ii) the YFCC 100 million dataset [104], (iii) Google's JFT 300 million dataset [82, 102], (iv) a subset of OpenImages [98], or (v) a subset of the full ImageNet dataset of 21,841 classes [11, 49, 99].

# 4 Main results

We now present our main experiments. First, we measure how much effective and relative robustness models achieve on the natural distribution shifts in our testbed. Then we investigate to what extent robustness on synthetic distribution shift is predictive of robustness on natural distribution shift.

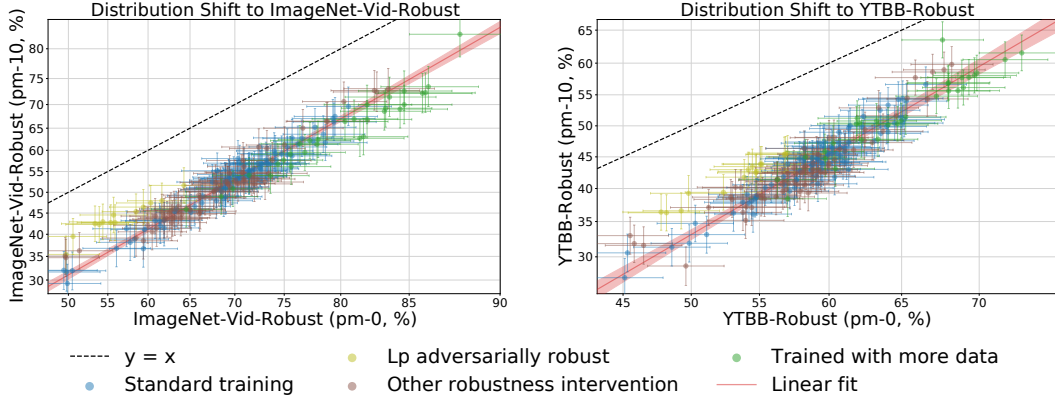

Figure 3: Model accuracies on the two consistency shifts: ImageNet-Vid-Robust (left), and YTBB-Robust (right). Both plots are shown with evaluation on pm-0 (anchor frames) on the x-axis and pm-10 (worst case prediction in a 20-frame neighborhood) on the y-axis. This plot shows that most current robustness interventions do not provide robustness to consistency distribution shifts. The notable outliers to this trend are $\ell_p$-adversarially robust models and EfficientNet-L2 (NoisyStudent). We color the adversarially robust models separately in this figure to illustrate this phenomenon. Confidence intervals, axis scaling, and the linear fit are computed similarly to Figure 2.

## 4.1   Results on natural distribution shifts

Following the categorization in Section 3, we measure the robustness of classification models on three types of natural distribution shift. Appendix J.2 contains variations of the figures referenced in this section. For further detail, we have made interactive plots available at http://robustness.imagenetv2.org/.

**Dataset shifts.** Figure 2 shows the effective robustness of models on the four dataset shifts in our testbed. In each case, we find that the standard test accuracy (x-axis) is a good predictor for the test accuracy under distribution shift (y-axis). The linear fit is best for ImageNetV2, ObjectNet, and ImageNet-Vid-Robust with respective $r^2$ scores of 1.00, 0.95, and 0.95, but is more noisy for YTBB-Robust ($r^2 = 0.83$). The noisy fit on YTBB-Robust is likely due to the fact that the categories in YTBB-Robust are not well aligned with those of ImageNet, where the models were trained [76]. Another potential reason is that the video test sets are significantly smaller (2,530 images in YTBB and 1,109 images in ImageNet-Vid-Robust).

In the high accuracy regime (above the 76% achieved by a ResNet-50), the main outliers in terms of positive effective robustness are models trained on substantially more data than the standard ImageNet training set. This includes a ResNet152 model trained on 11,000 ImageNet classes ($\rho = 2.1\%$) [99], several ResNeXt models trained on 1 billion images from Instagram ($\rho = 1.5\%$) [56], and the EfficientNet-L2 (NoisyStudent) model trained on a Google-internal JFT-300M dataset of 300 million images ($\rho = 1.1\%$) [102]. However, not all models trained on more data display positive effective robustness. For instance, a ResNet101 trained on the same JFT-300M dataset has an effective robustness of $\rho = -0.23\%$ [82]. We conduct additional experiments to investigate the effect of training data in Appendix B. Appendix H contains a full list of models with their effective robustness numbers. On YTBB-Robust, a few data augmentation strategies and $\ell_p$-robust models display positive effective robustness; we investigate this further in Appendix C.2.

**Consistency shifts.** We plot the effective robustness of models on consistency shifts in Figure 3. Interestingly, we observe that $\ell_p$-adversarially robust models display substantial effective robustness to ImageNet-Vid-Robust (average $\rho = 6.7\%$) and YTBB-Robust (average $\rho = 4.9\%$). This suggests that these models are not only more robust to synthetic perturbations, but also offer robustness for the perceptually small variations between consecutive video frames.

However, these gains in effective robustness do not necessarily lead to relative robustness. On average, relative robustness on both datasets is negative (average $\tau = -8.5\%$ on ImageNet-Vid-Robust and average $\tau = -0.7\%$ on YTBB-Robust for ResNet50 models). See Appendix C.2 (Figure 10) for a visual comparison. Among the models trained on more data, only one achieves both high accuracy and substantial effective robustness: EfficientNet-L2 (NoisyStudent) [102] has $\rho = 2.4\%$ and $\rho = 7.4\%$ on ImageNet-Vid-Robust and YTBB-Robust, respectively.

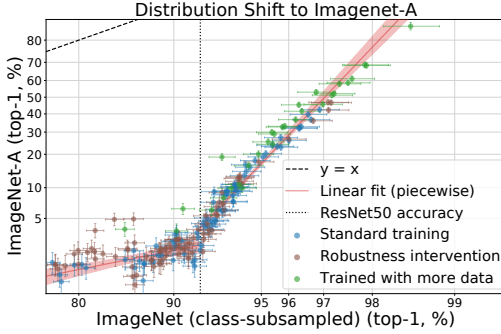

Figure 4: Model accuracies on ImageNet-A, a dataset adversarially filtered to contain only images incorrectly classified by a ResNet50 trained on ImageNet. This filtering results in a 'knee' curve: models with lower ImageNet accuracy than ResNet-50 have near-chance performance on ImageNet-A, while models with higher ImageNet accuracy improve drastically on ImageNet-A. The linear fit is computed piecewise around the ResNet50 model accuracy.

**Adversarially filtered shifts.** ImageNet-A [39] was created by classifying a set of images with a ResNet50 and only keeping the misclassified images. Interestingly, this approach creates a "knee" in the resulting scatter plot (see Figure 4): models below a ResNet50's standard accuracy have close to chance performance on ImageNet-A,[2] and models above a ResNet50's standard accuracy quickly close the accuracy gap. In the high accuracy regime, every percentage point improvement on ImageNet brings at least an 8% improvement on ImageNet-A. This is in contrast to datasets that are not constructed adversarially, where the initial accuracy drops are smaller, but later models make slow progress on closing the gap. These results demonstrate that adversarial filtering does not necessarily lead to harder distribution shifts.

## 4.2 Results on synthetic distribution shifts

Given the difficulty of collecting real world data to measure a model's robustness to natural distribution shifts, an important question is whether there are synthetic proxies. We now study to what extent robustness to the above synthetic distribution shifts predicts robustness on these natural distribution shifts.

In Figure 5, we analyze the predictiveness of one commonly studied synthetic robustness metric: average accuracy on image corruptions [38]. We compare this metric with effective robustness on ImageNetV2. While effective robustness is only one aspect (c.f. Section 2), it is a necessary prerequisite for a model to have helpful robustness properties.

The plots show that robustness under this synthetic distribution shift does not imply that the corresponding model has effective robustness on ImageNetV2 (the Pearson correlation coefficient is $r = 0.24$). In Appendix D.1, we repeat the above experiment for accuracy drop under PGD adversarial attacks [55] and also find a weak correlation ($r = -0.05$). Appendix D.2 further extends the experiment by comparing both synthetic distribution shift measures with the remaining natural distribution shifts in our testbed and reaches similar conclusions.

Our analysis of the aggregate measures proposed in prior work does not preclude that specific synthetic distribution shifts do predict behavior on natural distribution shifts. Instead, our results show that averaging a large number of synthetic corruptions does not yield a comprehensive robustness measure that also predicts robustness on natural distribution shift.

To extend on this analysis, in Appendix I we find that no individual synthetic measure in our testbed is a consistent predictor of natural distribution shift, but some synthetic shifts are substantially more predictive than others. For instance, $\ell_p$-robustness has the highest correlation with consistency shifts, and some image corruptions such as brightness or Gaussian blur have higher correlation with dataset shifts. However, our testbed indicates that these synthetic measures are not necessarily causal, i.e., models trained with brightness or Gaussian blur do not have substantial effective robustness on dataset shifts. Further analyzing relationships between individual synthetic and natural distribution shifts is an interesting avenue for future work.

## 4.3 Takeaways and discussion

To recap our results, we now discuss two of the central questions in our paper: Do current robustness interventions help on real data? And is synthetic robustness correlated with natural robustness?

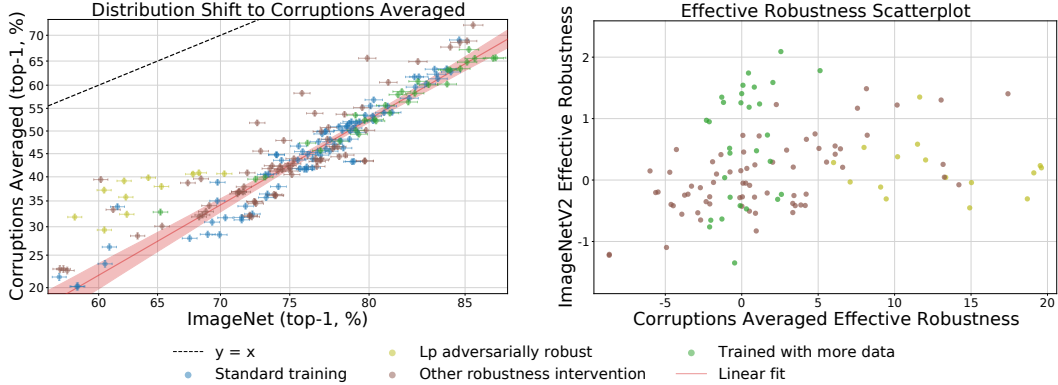

Figure 5: Model accuracies under image corruptions. Similar to Figure 2, the left plot shows the effective robustness for this synthetic distribution shift. Multiple non-standard models achieve substantial effective robustness, corroborating recent research progress on creating models robust to synthetic shifts. The right plot shows the correlation between the effective robustness for image corruptions and the ImageNetV2 distribution shift (top left in Figure 2) for the non-standard models. Image corruptions are only weakly predictive of effective robustness on ImageNetV2: there are several models that achieve high effective robustness under image corruptions but little to no effective robustness on ImageNetV2.

Across our study, current robustness interventions offer little to no improvement on the natural distribution shifts presently available.

For dataset shifts, we find that models trained with substantially more data yield a small improvement. However, the amount of extra data needed is orders of magnitude larger than the standard ImageNet training set, and the models show only small gains (in the best case improving the accuracy drop from 8.6% to 7.5% on ImageNetV2 for EfficientNet-L2 NoisyStudent). These results suggest that current robustness interventions methods do not provide benefits on the dataset shifts in our study.

For consistency shifts, adversarially trained models generally have effective robustness, but usually little or no relative robustness. On ImageNet-Vid-Robust, the baseline models without adversarial training still achieve higher accuracy under distribution shift. A notable outlier is EfficietNet-L2 (NoisyStudent) [102], which utilizes self-training and exhibits high effective robustness in the high accuracy regime. Self-training has recently been shown to help adversarial robustness as well [10, 61, 94]. Investigating the effect of self-training on robustness is an interesting direction for future work.

Moreover, we find that current aggregate metrics for synthetic robustness are at most weakly correlated with natural robustness. Effective robustness under non-adversarial image corruptions or $\ell_p$-attacks does not imply effective robustness to natural distribution shifts. While much progress has been made on creating models robust to synthetic distribution shift, new methods may be needed to handle natural shifts.

## 5 Related work

Our work is best seen as a unification of two independent lines of research—synthetic and natural distribution shift—not previously studied together. Synthetic distribution shifts have been studied extensively in the literature [28, 33, 38, 48, 57, 93]. We incorporate as many prior synthetic measures of robustness as possible. Our dataset largely confirms the high-level results from these papers (see Appendix E for additional discussion). For example, Ford et al. [31] provide evidence for the relationship between adversarial robustness and robustness to Gaussian noise. The study of natural distribution shifts has been an equally extensive research direction [2, 68, 76, 91]. When examining each natural distribution shift individually, we confirm the findings of earlier work that there is a consistent drop with a linear trend going from ImageNet to each of the other test sets [2, 68, 76].

We study the relationship between these two previously independent lines of work. By creating a testbed $100\times$ larger than prior work [27, 33, 47, 68], we are able to make several new observations. For instance, we show that robustness to synthetic distribution shift often behaves differently from robustness to natural distribution shift. We argue that it is important to control for accuracy when

measuring the efficacy of a robustness intervention. Viewed in this light, most interventions do not provide effective robustness. The main exception is training with more data, which improves robustness across natural distribution shifts. In some situations, $\ell_p$-adversarial robustness helps with natural distribution shift that asks for consistency across similar looking images.

Appendix L contains additional discussion of related work in more detail. Appendix L.1 broadly discusses the relationship of this work to other areas in machine learning. Appendix L.3 specifically revisits consistency shifts and explains why, in contrast to previous work [35], we find consistency robustness is only weakly correlated with color corruption robustness.

**Concurrent and subsequent work.** An early version of this paper with results on ImageNetV2 and ImageNet-Vid-Robust appeared on OpenReview in late 2019 [90]. Since then, two closely related papers have been published concurrently with the updated version of this paper.

Djolonga et al. [21] evaluate 40 models on the same natural distribution shifts as our paper. Our testbed is larger and contains 200 models with more robustness interventions. Overall both papers reach similar conclusions. Their focus is more on the connections to transfer learning while we focus more on comparisons between synthetic and natural distribution shifts. Djolonga et al. [21] also explore the performance of various models with a synthetic image dataset.

Hendrycks et al. [40] also study the connections between synthetic robustness and robustness to natural distribution shifts. This paper introduces a new dataset, ImageNet-R, that contains various renditions (sculptures, paintings, etc.) of 200 ImageNet classes as a new example of natural distribution shift. The paper then introduces DeepAugment, a new data augmentation technique based on synthetic image transformations, and find that this robustness intervention is effective on ImageNet-R. In Appendix L.2, we analyze the ImageNet-R test set and DeepAugment models, as well as the closely related ImageNet-Sketch test set [95], in more detail.

At a high-level, ImageNet-R and ImageNet-Sketch follow the trends of the other dataset shifts in our testbed, with models trained on extra data providing the most robustness (up to $\rho = 29.1\%$ on ImageNet-R, though the effect is not uniform, similar to the other dataset shifts). After the models trained on more data, we find that DeepAugment (in combination with AugMix [41]) achieves substantial effective robustness ($\rho = 11.2\%$). Interestingly, adversarial robustness also leads to effective robustness on ImageNet-R. An AdvProp model [100] achieves the highest absolute accuracy on ImageNet-R for a model trained without extra data (57.8%) and has effective robustness $\rho = 7.5\%$. A model with feature denoising and trained with PGD-style robust optimization [55, 101] achieves the highest effective robustness on ImageNet-R ($\rho = 22.7\%$) and also positive relative robustness ($\tau = 5.7\%$).

# 6 Conclusion

The goal of robust machine learning is to develop methods that function reliably in a wide variety of settings. So far, this research direction has focused mainly on synthetic perturbations of existing test sets, highlighting important failure cases and initiating progress towards more robust models. Ultimately, the hope is that the resulting techniques also provide benefits on real data. Our paper takes a step in this direction and complements the current synthetic robustness tests with comprehensive experiments on distribution shifts arising from real data.

We find that current image classification models still suffer from substantial accuracy drops on natural distribution shifts. Moreover, current robustness interventions – while effective against synthetic perturbations – yield little to no consistent improvements on real data. The only approach providing broad benefits is training on larger datasets, but the gains are small and inconsistent.

Overall, our results show a clear challenge for future research. Even training on 1,000 times more data is far from closing the accuracy gaps, so robustness on real data will likely require new algorithmic ideas and better understanding of how training data affects robustness. Our results indicate two immediate steps for work in this area: robustness metrics should control for baseline accuracy, and robust models should additionally be evaluated on natural distribution shifts. We hope that our comprehensive testbed with nuanced robustness metrics and multiple types of distribution shift will provide a clear indicator of progress on the path towards reliable machine learning on real data.

## Broader Impact

Robustness is one of the key problems that prevents deploying machine learning in the real world and harnessing the associated benefits. A canonical example is image classification for medical diagnosis. As was found when researchers attempted to deploy a neural network to detect diabetes from retina images, "an accuracy assessment from a lab goes only so far. It says nothing of how the AI will perform in the chaos of a real-world environment" [3]. Similarly, researcher also found that current methods for chest X-ray classification are brittle even in the absence of recognized confounders [110]. If models were robust, then this transfer to the real world would be straightforward. Unfortunately, achieving robustness on real data is still a substantial challenge for machine learning.

Our work studies how robust current image classification methods are to distribution shifts arising in real data. We hope that our paper will have a positive effect on the study of distribution shifts and allow researchers to more accurately evaluate to what extent a proposed technique increases the robustness to particular forms of distribution shift. This will allow researchers to better understand how a deployed system will work in practice, without actually having to deploy it first and users potentially suffering negative consequences.

However, there are several potential ways in which our study could cause unintended harm. It is possible that our paper might be used as an argument to stop performing research on some synthetic forms of robustness, e.g., adversarial examples or common corruptions. This is not our intention. These forms of corruption are interesting independent of any correlation to existing natural distribution shift (e.g., adversarial examples are a genuine security problem).

We only capture a small number of natural distribution shifts among all the possible distribution shifts. We selected these shifts because they have been used extensively in the literature and are concrete examples of the types of distribution shift we would like models to be robust to. It is likely that there are shifts that we do not capture, and so even if the shifts we define were to be completely solved, other shifts would remain a concern.

One significant form of distribution shift we do not evaluate is dataset bias in representing different demographic groups. For example, the Inclusive Images dataset [75] attempts to correct for the geographical bias introduced in the Open Images dataset [51] by including a more balanced representation of images from Africa, Asia, and South America. Neglecting such implicit biases in the data distribution can harm underrepresented demographic groups. Ultimately, evaluating on fixed datasets may not be enough, and validating the fairness and safety of deployable machine learning requires careful analysis in the application domain.

Finally, more reliable machine learning can also enable negative uses cases, e.g., widespread surveillance or autonomous weapon systems. As with many technologies, these risks require careful regulation and awareness of unintended consequences arising from technological advances.

## Acknowledgments and Disclosure of Funding

We would like to thank Logan Engstrom, Justin Gilmer, Moritz Hardt, Daniel Kang, Jerry Li, Percy Liang, Nelson Liu, John Miller, Preetum Nakkiran, Rebecca Roelofs, Aman Sinha, Jacob Steinhardt, and Dimitris Tsipras for helpful conversations while working on this paper.

This research was generously supported in part by ONR awards N00014-17-1-2191, N00014-17-1-2401, and N00014-18-1-2833, the DARPA Assured Autonomy (FA8750-18-C-0101) and Lagrange (W911NF-16-1-0552) programs, a Siemens Futuremakers Fellowship, an Amazon AWS AI Research Award.

## Footnotes

[1]For ObjectNet [2], Borji [7] has pointed out potential label quality issues, but also found that a substantial accuracy drop remains when taking these issues into account.

[2]Chance performance is 0.5% as ImageNet-A contains 200 classes.

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
