[Supplementary Material]

# A    Testbed overview

Figure 6: An overview of our testbed. Each row is a model, and each column is an evaluation setting. For the corruptions, we display each of the five severities defined in [38]. We also plot in-memory and on-disk versions of each corruption as jpeg compression was found to be a confounding factor in [31]. A few cells are empty due to resource constraints. Testbed code and data is provided at https://modestyachts.github.io/imagenet-testbed/.

# B How does the amount of training data impact robustness?

As discussed in Section 4.1, multiple models trained on more data achieve positive effective robustness on dataset shifts. However, this effect is not uniform. Among others, the ResNet101 model trained on JFT-300M has negligible effective robustness ($\rho = -0.23\%$) despite being trained on $300\times$ more data than standard ImageNet models. A possible explanation is that differences in label diversity or quality play a role in promoting robustness. We investigate the role of data in more detail with two experiments.

**Varying the number of images per class.** We start by subsampling the ILSVRC-2012 training set by factors of $\{2, 4, 8, 16, 32\}$ and show the impact on accuracy and robustness on ImageNetV2 in Figure 7. While larger training subsets yield higher accuracies, they do not improve effective robustness, at least for ImageNetV2.

**Varying the number of classes.** Next, we subsample ImageNet in a more biased way by varying the set of classes. First, we create three subsets of the ILSVRC training set with 500, 250, and 125 classes and train models on these subsets. We then evaluate all models on the 125 class subset and show the results in Figure 7. Varying the number of classes again affects accuracies, but does not impact effective robustness.

Figure 7: To investigate the impact of training data on robustness, we vary the training data along two axes: the number of images per class (left), and the number of classes (right). Although models trained on more data provide improvements in effective robustness, we find that subsampling the training set has no impact on effective robustness. Confidence intervals, axis scaling, and the linear fit are computed similarly to Figure 2.

Our experiments suggest that neither growing the number of images nor classes in an i.i.d. fashion are effective robustness interventions. Nevertheless, Figure 2 shows that larger datasets can provide meaningful robustness improvements. This disparity may be due to limitations of emulating dataset growth by subsampling ILSVRC. For one, our experiments consider only i.i.d. subsets of the training images or classes. Another possibility is that increases in dataset size may only improve robustness after the dataset is large enough so that the accuracy on the original distribution is nearly saturated. Our experiments only observe dataset sizes smaller than ILSVRC, which may fall below this inflection point. Studying the effect of data on robustness is an important direction for future work.

# C  Relative and effective robustness

## C.1  Relative and effective robustness graphical sketch

A central question we address in our paper is whether current methodologies provide meaningful robustness to natural distribution shifts. We discuss how both relative robustness and effective robustness are needed to disentangle the confounding effect of original model accuracy. In Figure 8, we graphically illustrate this notion of relative robustness.

Figure 8: While a hypothetical intervention (green), applied to a baseline model (blue), leads to effective robustness (it is above the red line), it reduces the model's accuracy under distribution shift. Hence it fails to provide *relative* robustness. An ideal intervention would place the model in the white quadrant - positive effective and relative robustness.

## C.2  Relative and effective robustness for ResNet50 models

We provide additional plots depicting a subset of the models in our testbed. In order to make an equal comparison, we only plot ResNeet50 variants, models which slightly modify the training data or architecture of a base ResNet50. The plots in this section thus describe what the relative and effective robustness properties of various robustness interventions look like on a standard ResNet50. The models can be directly compared with each other since the base model before intervention is the same.

For natural dataset shifts, the plots in Figure 9 demonstrate that the only models that have consistently positive relative and positive effective robustness are models that are trained on more data. However, the effect is small, and not all models trained on more data are more robust. On YTBB-Robust specifically, a few data augmentation strategies from ImageNet-C provide significant both effective and relative robustness: training on greyscale ($\rho = 6.9\%$, $\tau = 1.8\%$); training on pixelate ($\rho = 5.4\%$, $\tau = 2.0\%$); training on jpeg compression ($\rho = 5.4\%$, $\tau = 6.3\%$); training on gaussian noise, contrast, motion blur, and jpeg compression ($\rho = 4.8\%$, $\tau = 5.0\%$); and training on gaussian noise ($\rho = 3.6\%$, $\tau = 4.0\%$). However, this performance is not consistent across the natural distribution shifts. Exploring why these data augmentation strategies are helpful on YTBB-Robust is an interesting direction for future work. Additionally, while some $\ell_p$-adversarially robust models display significant effective robustness on YTBB-Robust - $\ell_2$ robust ResNet50 ($\rho = 6.4\%$), $\ell_{\inf}$ robust ResNet50 ($\rho = 6.4\%$), and ResNet50 smoothed with 0.25 gaussian noise and adversarially 1-step PGD trained ($\rho = 5.0$) - in most cases, they fail to provide positive relative robustness.

For natural consistency shifts, the plots in Figure 10 demonstrate that while adversarially robust models provide effective robustness (average $\rho = 4.3\%$ on ImageNet-Vid-Robust and average $\rho = 3.9\%$ on YTBB-Robust), they only sometimes provide relative robustness on YTBB-Robust.

For the adversarially filtered shift, the plot in Figure 11 demonstrates that robustness interventions have little impact on ImageNet-A accuracy. Most of the "knee"-like response curve can be explained as an artifact of the adversarial filtering, with the knee occuring at the ResNet50 model accuracy.

Figure 9: Relative and effective robustness for models that are variants of a ResNet50. Model accuracies are displayed on the four natural dataset shifts: ImageNetV2 (top left), ObjectNet (top right), ImageNet-Vid-Robust-anchor (bottom left), and YTBB-Robust-anchor (bottom right). These plots demonstrate that the only models that have consistently positive relative and positive effective robustness are models that are trained on more data. However, the effect is small, and not all models trained on more data are more robust. Confidence intervals, axis scaling, and the linear fit are computed similarly to Figure 2.

Figure 10: Relative and effective robustness for models that are variants of a ResNet50. Model accuracies are displayed the two consistency shifts: ImageNet-Vid-Robust (left), and YTBB-Robust (right). These plots demonstrate that while adversarially robust models provide effective robustness, they do not necessarily provide relative robustness. Confidence intervals, axis scaling, and the linear fit are computed similarly to Figure 2.

Figure 11: Relative and effective robustness for models that are variants of a ResNet50. Model accuracies are displayed on ImageNet-A, a dataset adversarially filtered to contain only images incorrectly classified by a ResNet50 trained on ImageNet. Due to the "knee"-like response curve, an artifact of the adversarial filtering, effective robustness is defined piecewise around the ResNet50 model accuracy point. The plot demonstrates that robustness interventions have little impact on ImageNet-A accuracy. However, the effect is small, and not all models trained on more data are more robust. Confidence intervals, axis scaling, and the linear fit are computed similarly to Figure 2.

# D Synthetic vs. natural robustness

## D.1 Adversarial attacks vs. ImageNetV2

In Figure 12, we analyze the predictiveness of accuracy under $\ell_p$ adversarial attacks and compare this metric with effective robustness on ImageNetV2. This plot is similar to Figure 5, but analyzes $\ell_p$ attacks instead of image corruptions. The plots show that robustness under $\ell_p$ attacks does not imply that the corresponding model has effective robustness on ImageNetV2 (the Pearson correlation coefficient is $r = -0.05$).

Figure 12: Model accuracies under $\ell_p$ adversarial PGD attacks. Similar to Figure 2, the left plot shows the effective robustness for this synthetic distribution shift. Multiple non-standard models achieve substantial effective robustness, corroborating recent research progress on creating adversarially robust models. The right plot shows the correlation between the effective robustness for $\ell_p$ attacks and the ImageNetV2 distribution shift (top left in Figure 2) for the non-standard models. $\ell_p$ attacks are only weakly predictive of effective robustness on ImageNetV2: there are several models that achieve high effective robustness under $\ell_p$ but little to no effective robustness on ImageNetV2.

## D.2 Effective robustness scatterplots

In this section, we further explore to what extent robustness to synthetic distribution shifts predicts robustness on natural distribution shift. We extend the analysis in Figures 5 and 12 by computing effective robustness on all natural distribution shifts and comparingn them against effective robustness on synthetic distribution shifts.

For natural dataset shifts, the scatter plots in Figure 13 are weakly correlated (the Pearson correlation coefficients are $r = 0.24, -0.05, -0.01, -0.26, 0.61, 0.30, 0.52, 0.36$ in reading order), indicating that improved robustness to corruptions or adversarial attacks in general does not improve effective robustness under natural dataset shifts. Of the group, the two strongest correlations are effective robustness between ImageNet-Vid-Robust and image corruptions ($r = 0.61$) and between YTBB-Robust and image corruptions ($r = 0.52$). While not very strong, the correlations are significant, and exploring this phenomenon between image corruptions and video anchor frames is an interesting direction for future work.

For natural consistency shifts, the plots in Figure 14 are largely uncorrelated, with the exception that accuracy on adversarial attacks is correlated with effective robustness on consistency shifts for lp adversarially models. However, as explored in Appendix C.2, effective robustness on these shifts does not always imply relative robustness.

For the adversarially filtered shift, as seen in Figure 15, after computing effective robustness piecewise around the ResNet50 accuracy, there is no observed correlation between the synthetic and natural robustness measures on ImageNet-A.

Figure 13: We compare the effective robustness of models with their accuracy drop due to corruptions (left column) and adversarial attacks (right column). The effective robustness is computed with respect to linear fits on the four natural dataset shifts: ImageNetV2 (first row), ObjectNet (second row), ImageNet-Vid-Robust-anchor (third row), and YTBB-Robust-anchor (fourth row). The measures are largely uncorrelated, indicating that improved robustness to corruptions or adversarial attacks does not improve effective robustness under natural dataset shifts.

Figure 14: We compare the effective robustness of models with their accuracy drop due to corruptions (left column) and adversarial attacks (right column). The effective robustness is computed with respect to linear fits on the two consistency shifts: ImageNet-Vid-Robust (first row), and YTBB-Robust (second row). The measures are largely uncorrelated, with the exception that accuracy on adversarial attacks is correlated with effective robustness on consistency shifts for lp adversarially robust models.

Figure 15: We compare the effective robustness of models with their accuracy drop due to corruptions (left column) and adversarial attacks (right column). The effective robustness is computed with respect to a linear fit on ImageNet-A, the adversarially filtered shift. After computing effective robustness piecewise around the ResNet50 accuracy, there is no observed correlation between the synthetic and natural robustness measures.

# E    Corruption robustness

Figure 16: A detailed view of corruption robustness, with cells sampled from the main grid in Figure 6. Here we present ResNet50s trained on some of the corruptions from the ImageNet-C benchmark, as well as the best model trained on more data, FixResNeXt101_32x48d_v2, and the best model trained on just the standard training set, efficientnet-b8-advprop-autoaug.

We have already seen that corruption robustness does not promote effective robustness, or robustness to real distribution shift. Here, we analyze whether robustness to some corruptions transfers to others, and what may contribute to corruption robustness. Figure 16 shows the result of training various ResNet50s[3] on a few corruptions from ImageNet-C.

In line with prior work, this plot here tells us that training against one type of synthetic corruption or one set of synthetic corruption does not transfer well to other corruptions. There are cases where transfer does happen, but overall the models are only robust to the corruption they are trained on.

It is also interesting to note (from Figure 6) that PGD models actually see a drop in robustness to low frequency corruptions such as contrast, a phenomenon also observed in [107].

# F    Evaluation settings in the testbed

## F.1    Natural distribution shifts

For ImageNetV2, we evaluate on the following datasets: imagenetv2-matched-frequency, imagenetv2-matched-frequency-format-val, imagenetv2-threshold-0.7, imagenetv2-threshold-0.7-format-val, imagenetv2-top-images, imagenetv2-top-images-format-val. The format-val versions are variants of the original dataset encoded with jpeg settings similar to the original one. Unless otherwise stated, results in our paper referring to imagenetv2 are for imagenetv2-matched-frequency-format-val.

For ObjectNet, we obtained a beta version of the dataset through personal correspondance. Each image in the dataset was then cropped by 2px on each side following the authors' instructions. Predictions were taken over only the classes that also appeared in the 1000 classes for the ImageNet validation set.

For ImageNet-Vid-Robust and YTBB-Robust, we look at the anchor frames in the dataset and evaluate the benign accuracy for pm0. For pm10, we look at up to 20 nearest frames marked "similar" to the anchor frame in the dataset and count it as a misclassification if any one of the predictions is wrong.

For ImageNet-A, predictions were taken over only the classes that also appeared in the 1000 classes for the ImageNet validation set.

## F.2    Corruptions

We include 38 different corruption types: greyscale (in memory), gaussian noise (in memory and on disk), shot noise (in memory and on disk), impulse noise (in memory and on disk), speckle noise (in memory and on disk), gaussian blur (in memory and on disk), defocus blur (in memory and on disk), glass blur (on disk), motion blur (in memory and on disk), zoom blur (in memory and on disk), snow (in memory and on disk), frost (in memory and on disk), fog (in memory and on disk), spatter (in memory and on disk), brightness (in memory and on disk), contrast (in memory and on disk), saturate (in memory and on disk), pixelate (in memory and on disk), jpeg compression (in memory and on disk), elastic transform (in memory and on disk).

For each corruption, we average over the five severities.

We make sure to make the distinction between in memory corruptions, for which we provide custom fast gpu implementations, and on disk corruptions, for which we use the publicly available ImageNet-C dataset, since it was reported in [31] that jpeg compression can have a significant impact on model accuracies (indeed, as evidenced by Figure 16).

## F.3    Adversarial attacks

We run the following 4 pgd attacks one each model with these settings:

`pgd.linf.eps0.5` Norm: 0.5/255, Step size: 5.88e-5, Num steps: 100

`pgd.linf.eps2` Norm: 2/255, Step size: 2.35e-4, Num steps: 100

`pgd.l2.eps0.1` Norm: 0.1, Step size: 0.01, Num steps: 100

`pgd.l2.eps0.5` Norm: 0.5, Step size: 0.05, Num steps: 100

Most of the models were attacked with only 10% of the dataset (in a class-balanced manner) due to computational constraints. These models are displayed with larger error bars in the plots.

## F.4    Stylized Imagenet

We use the stylized imagenet dataset used by [34] as another evaluation dataset.

## F.5    125 class evaluation

For the 125 subsampled class evaluation, we evaluate on the following classes from ILSVRC:

```
n01494475 n01630670 n01644373 n01644900 n01669191 n01677366 n01697457
n01742172 n01796340 n01829413 n01871265 n01924916 n01944390 n01978287
n01980166 n02007558 n02009229 n02017213 n02033041 n02037110 n02056570
n02071294 n02085936 n02086079 n02093428 n02093991 n02095314 n02095570
n02096294 n02096437 n02097474 n02100236 n02100583 n02102318 n02105056
n02107574 n02112706 n02113023 n02114855 n02128925 n02134418 n02138441
```

```
n02165105 n02219486 n02226429 n02264363 n02280649 n02441942 n02483708
n02486261 n02488291 n02492035 n02641379 n02730930 n02777292 n02790996
n02795169 n02808440 n02814533 n02814860 n02837789 n02859443 n02892201
n02895154 n02948072 n02951585 n02977058 n03000247 n03110669 n03201208
n03208938 n03216828 n03240683 n03250847 n03272562 n03297495 n03337140
n03376595 n03379051 n03447721 n03492542 n03527444 n03535780 n03642806
n03670208 n03673027 n03692522 n03710193 n03775071 n03832673 n03838899
n03840681 n03868242 n03873416 n03877845 n03884397 n03908714 n03920288
n03933933 n04004767 n04009552 n04037443 n04041544 n04067472 n04074963
n04099969 n04125021 n04141975 n04149813 n04204238 n04208210 n04229816
n04266014 n04310018 n04330267 n04335435 n04336792 n04355338 n04417672
n04479046 n04505470 n07715103 n07875152 n09256479 n12620546
```

# G Models in the testbed

The following list contains all models we evaluated on ImageNet with references and links to the corresponding source code. Also noted is the model type used to color the plots in the paper.

1. BiT-M-R50x1-ILSVRC2012 [49]. Trained with more data model. `https://github.com/google-research/big_transfer`

2. BiT-M-R50x3-ILSVRC2012 [49]. Trained with more data model. `https://github.com/google-research/big_transfer`

3. BiT-M-R101x1-ILSVRC2012 [49]. Trained with more data model. `https://github.com/google-research/big_transfer`

4. BiT-M-R101x3-ILSVRC2012 [49]. Trained with more data model. `https://github.com/google-research/big_transfer`

5. BiT-M-R152x4-ILSVRC2012 [49]. Trained with more data model. `https://github.com/google-research/big_transfer`

6. FixPNASNet [92]. Standard training model. `https://github.com/facebookresearch/FixRes`

7. FixResNeXt101_32x48d [92]. Trained with more data model. `https://github.com/facebookresearch/FixRes`

8. FixResNeXt101_32x48d_v2 [92]. Trained with more data model. `https://github.com/facebookresearch/FixRes`

9. FixResNet50 [92]. Standard training model. `https://github.com/facebookresearch/FixRes`

10. FixResNet50CutMix [92]. Robustness intervention model. `https://github.com/facebookresearch/FixRes`

11. FixResNet50CutMix_v2 [92]. Robustness intervention model. `https://github.com/facebookresearch/FixRes`

12. FixResNet50_no_adaptation [92]. Standard training model. `https://github.com/facebookresearch/FixRes`

13. FixResNet50_v2 [92]. Standard training model. `https://github.com/facebookresearch/FixRes`

14. alexnet [50]. Standard training model. `https://github.com/Cadene/pretrained-models.pytorch`

15. alexnet_lpf2 [115]. Robustness intervention model. `https://github.com/adobe/antialiased-cnns`

16. alexnet_lpf3 [115]. Robustness intervention model. `https://github.com/adobe/antialiased-cnns`

17. alexnet_lpf5 [115]. Robustness intervention model. `https://github.com/adobe/antialiased-cnns`

18. bninception [46]. Standard training model. `https://github.com/Cadene/pretrained-models.pytorch`

19. bninception-imagenet21k [46]. Trained with more data model. `https://github.com/dmlc/mxnet-model-gallery/blob/master/imagenet-21k-inception.md`

20. cafferesnet101 [37]. Standard training model. `https://github.com/Cadene/pretrained-models.pytorch`

21. densenet121 [43]. Standard training model. `https://github.com/Cadene/pretrained-models.pytorch`

22. densenet121_lpf2 [115]. Robustness intervention model. `https://github.com/adobe/antialiased-cnns`

23. densenet121_lpf3 [115]. Robustness intervention model. `https://github.com/adobe/antialiased-cnns`

24. densenet121_lpf5 [115]. Robustness intervention model. `https://github.com/adobe/antialiased-cnns`

25. densenet161 [43]. Standard training model. `https://github.com/Cadene/pretrained-models.pytorch`

26. densenet169 [43]. Standard training model. `https://github.com/Cadene/pretrained-models.pytorch`

27. densenet201 [43]. Standard training model. `https://github.com/Cadene/pretrained-models.pytorch`

28. dpn107 [11]. Trained with more data model. `https://github.com/Cadene/pretrained-models.pytorch`

29. dpn131 [11]. Standard training model. `https://github.com/Cadene/pretrained-models.pytorch`

30. dpn68 [11]. Standard training model. `https://github.com/Cadene/pretrained-models.pytorch`

31. dpn68b [11]. Trained with more data model. `https://github.com/Cadene/pretrained-models.pytorch`

32. dpn92 [11]. Trained with more data model. `https://github.com/Cadene/pretrained-models.pytorch`

33. dpn98 [11]. Standard training model. `https://github.com/Cadene/pretrained-models.pytorch`

34. efficientnet-b0 [88]. Standard training model. `https://github.com/tensorflow/tpu/tree/master/models/official/efficientnet`

35. efficientnet-b0-advprop-autoaug [100]. Robustness intervention model. `https://github.com/tensorflow/tpu/tree/master/models/official/efficientnet`

36. efficientnet-b0-autoaug [15]. Standard training model. `https://github.com/tensorflow/tpu/tree/master/models/official/efficientnet`

37. efficientnet-b1 [88]. Standard training model. `https://github.com/tensorflow/tpu/tree/master/models/official/efficientnet`

38. efficientnet-b1-advprop-autoaug [100]. Robustness intervention model. `https://github.com/tensorflow/tpu/tree/master/models/official/efficientnet`

39. efficientnet-b1-autoaug [15]. Standard training model. `https://github.com/tensorflow/tpu/tree/master/models/official/efficientnet`

40. efficientnet-b2 [88]. Standard training model. `https://github.com/tensorflow/tpu/tree/master/models/official/efficientnet`

41. efficientnet-b2-advprop-autoaug [100]. Robustness intervention model. `https://github.com/tensorflow/tpu/tree/master/models/official/efficientnet`

42. efficientnet-b2-autoaug [15]. Standard training model. `https://github.com/tensorflow/tpu/tree/master/models/official/efficientnet`

43. efficientnet-b3 [88]. Standard training model. `https://github.com/tensorflow/tpu/tree/master/models/official/efficientnet`

44. efficientnet-b3-advprop-autoaug [100]. Robustness intervention model. `https://github.com/tensorflow/tpu/tree/master/models/official/efficientnet`

45. efficientnet-b3-autoaug [15]. Standard training model. `https://github.com/tensorflow/tpu/tree/master/models/official/efficientnet`

46. efficientnet-b4 [88]. Standard training model. `https://github.com/tensorflow/tpu/tree/master/models/official/efficientnet`

47. efficientnet-b4-advprop-autoaug [100]. Robustness intervention model. `https://github.com/tensorflow/tpu/tree/master/models/official/efficientnet`

48. efficientnet-b4-autoaug [15]. Standard training model. `https://github.com/tensorflow/tpu/tree/master/models/official/efficientnet`

49. efficientnet-b5 [88]. Standard training model. `https://github.com/tensorflow/tpu/tree/master/models/official/efficientnet`

50. efficientnet-b5-advprop-autoaug [100]. Robustness intervention model. `https://github.com/ten sorflow/tpu/tree/master/models/official/efficientnet`

51. efficientnet-b5-autoaug [15]. Standard training model. `https://github.com/tensorflow/tpu/ tree/master/models/official/efficientnet`

52. efficientnet-b5-randaug [16]. Standard training model. `https://github.com/tensorflow/tpu/ tree/master/models/official/efficientnet`

53. efficientnet-b6-advprop-autoaug [100]. Robustness intervention model. `https://github.com/ten sorflow/tpu/tree/master/models/official/efficientnet`

54. efficientnet-b6-autoaug [15]. Standard training model. `https://github.com/tensorflow/tpu/ tree/master/models/official/efficientnet`

55. efficientnet-b7-advprop-autoaug [100]. Robustness intervention model. `https://github.com/ten sorflow/tpu/tree/master/models/official/efficientnet`

56. efficientnet-b7-autoaug [15]. Standard training model. `https://github.com/tensorflow/tpu/ tree/master/models/official/efficientnet`

57. efficientnet-b7-randaug [16]. Standard training model. `https://github.com/tensorflow/tpu/ tree/master/models/official/efficientnet`

58. efficientnet-b8-advprop-autoaug [100]. Robustness intervention model. `https://github.com/ten sorflow/tpu/tree/master/models/official/efficientnet`

59. efficientnet-l2-noisystudent [102]. Trained with more data model. `https://github.com/rwightm an/pytorch-image-models`

60. facebook_adv_trained_resnet152_baseline [101]. Robustness intervention model. `https://github .com/facebookresearch/ImageNet-Adversarial-Training`

61. facebook_adv_trained_resnet152_denoise [101]. Robustness intervention model. `https://github .com/facebookresearch/ImageNet-Adversarial-Training`

62. facebook_adv_trained_resnext101_denoiseAll [101]. Robustness intervention model. `https://gith ub.com/facebookresearch/ImageNet-Adversarial-Training`

63. fbresnet152 [37]. Standard training model. `https://github.com/Cadene/pretrained-models. pytorch`

64. google_resnet101_jft-300M [82]. Trained with more data model.

65. googlenet/inceptionv1 [85]. Standard training model. `https://github.com/pytorch/vision/tr ee/master/torchvision/models`

66. inceptionresnetv2 [37]. Standard training model. `https://github.com/Cadene/pretrained-m odels.pytorch`

67. inceptionv3 [86]. Standard training model. `https://github.com/Cadene/pretrained-models. pytorch`

68. inceptionv4 [87]. Standard training model. `https://github.com/Cadene/pretrained-models. pytorch`

69. instagram-resnext101_32x16d [56]. Trained with more data model. `https://github.com/faceb ookresearch/WSL-Images`

70. instagram-resnext101_32x32d [56]. Trained with more data model. `https://github.com/faceb ookresearch/WSL-Images`

71. instagram-resnext101_32x48d [56]. Trained with more data model. `https://github.com/faceb ookresearch/WSL-Images`

72. instagram-resnext101_32x8d [56]. Trained with more data model. `https://github.com/faceboo kresearch/WSL-Images`

73. mnasnet0_5 [89]. Standard training model. `https://github.com/pytorch/vision/tree/mas ter/torchvision/models`

74. mnasnet1_0 [89]. Standard training model. `https://github.com/pytorch/vision/tree/mas ter/torchvision/models`

75. mobilenet_v2 [73]. Standard training model. `https://github.com/pytorch/vision/tree/mas ter/torchvision/models`

76. mobilenet_v2_lpf2 [115]. Robustness intervention model. `https://github.com/adobe/antiali`
`ased-cnns`

77. mobilenet_v2_lpf3 [115]. Robustness intervention model. `https://github.com/adobe/antiali`
`ased-cnns`

78. mobilenet_v2_lpf5 [115]. Robustness intervention model. `https://github.com/adobe/antiali`
`ased-cnns`

79. nasnetalarge [117]. Standard training model. `https://github.com/Cadene/pretrained-model`
`s.pytorch`

80. nasnetamobile [117]. Standard training model. `https://github.com/Cadene/pretrained-mod`
`els.pytorch`

81. pnasnet5large [53]. Standard training model. `https://github.com/Cadene/pretrained-model`
`s.pytorch`

82. polynet [116]. Standard training model. `https://github.com/Cadene/pretrained-models.py`
`torch`

83. resnet101 [37]. Standard training model. `https://github.com/Cadene/pretrained-models.`
`pytorch`

84. resnet101-tencent-ml-images [97]. Trained with more data model. `https://github.com/Tencent`
`/tencent-ml-images`

85. resnet101_cutmix [108]. Robustness intervention model. `https://github.com/clovaai/CutMi`
`x-PyTorch`

86. resnet101_lpf2 [115]. Robustness intervention model. `https://github.com/adobe/antialias`
`ed-cnns`

87. resnet101_lpf3 [115]. Robustness intervention model. `https://github.com/adobe/antialias`
`ed-cnns`

88. resnet101_lpf5 [115]. Robustness intervention model. `https://github.com/adobe/antialias`
`ed-cnns`

89. resnet152 [37]. Standard training model. `https://github.com/Cadene/pretrained-models.`
`pytorch`

90. resnet152-imagenet11k [99]. Trained with more data model. `https://github.com/tornadomeet`
`/ResNet`

91. resnet18 [37]. Standard training model. `https://github.com/Cadene/pretrained-models.py`
`torch`

92. resnet18-rotation-nocrop_40 [28]. Robustness intervention model. `https://github.com/MadryLa`
`b/spatial-pytorch`

93. resnet18-rotation-random_30 [28]. Robustness intervention model. `https://github.com/Madry`
`Lab/spatial-pytorch`

94. resnet18-rotation-random_40 [28]. Robustness intervention model. `https://github.com/Madry`
`Lab/spatial-pytorch`

95. resnet18-rotation-standard_40 [28]. Robustness intervention model. `https://github.com/Madry`
`Lab/spatial-pytorch`

96. resnet18-rotation-worst10_30 [28]. Robustness intervention model. `https://github.com/Madry`
`Lab/spatial-pytorch`

97. resnet18-rotation-worst10_40 [28]. Robustness intervention model. `https://github.com/Madry`
`Lab/spatial-pytorch`

98. resnet18_lpf2 [115]. Robustness intervention model. `https://github.com/adobe/antialiased`
`-cnns`

99. resnet18_lpf3 [115]. Robustness intervention model. `https://github.com/adobe/antialiased`
`-cnns`

100. resnet18_lpf5 [115]. Robustness intervention model. `https://github.com/adobe/antialiased`
`-cnns`

101. resnet18_ssl [104]. Trained with more data model. `https://github.com/facebookresearch/se mi-supervised-ImageNet1K-models`

102. resnet18_swsl [104]. Trained with more data model. `https://github.com/facebookresearch/ semi-supervised-ImageNet1K-models`

103. resnet34 [37]. Standard training model. `https://github.com/Cadene/pretrained-models.py torch`

104. resnet34_lpf2 [115]. Robustness intervention model. `https://github.com/adobe/antialiased -cnns`

105. resnet34_lpf3 [115]. Robustness intervention model. `https://github.com/adobe/antialiased -cnns`

106. resnet34_lpf5 [115]. Robustness intervention model. `https://github.com/adobe/antialiased -cnns`

107. resnet50 [37]. Standard training model. `https://github.com/Cadene/pretrained-models.py torch`

108. resnet50-randomized_smoothing_noise_0.00 [13]. Standard training model. `https://github.com /locuslab/smoothing`

109. resnet50-randomized_smoothing_noise_0.25 [13]. Robustness intervention model. `https://github .com/locuslab/smoothing`

110. resnet50-randomized_smoothing_noise_0.50 [13]. Robustness intervention model. `https://github .com/locuslab/smoothing`

111. resnet50-randomized_smoothing_noise_1.00 [13]. Robustness intervention model. `https://github .com/locuslab/smoothing`

112. resnet50-smoothing_adversarial_DNN_2steps_eps_512_noise_0.25 [72]. Robustness intervention model. `https://github.com/Hadisalman/smoothing-adversarial`

113. resnet50-smoothing_adversarial_DNN_2steps_eps_512_noise_0.50 [72]. Robustness intervention model. `https://github.com/Hadisalman/smoothing-adversarial`

114. resnet50-smoothing_adversarial_DNN_2steps_eps_512_noise_1.00 [72]. Robustness intervention model. `https://github.com/Hadisalman/smoothing-adversarial`

115. resnet50-smoothing_adversarial_PGD_1step_eps_512_noise_0.25 [72]. Robustness intervention model. `https://github.com/Hadisalman/smoothing-adversarial`

116. resnet50-smoothing_adversarial_PGD_1step_eps_512_noise_0.50 [72]. Robustness intervention model. `https://github.com/Hadisalman/smoothing-adversarial`

117. resnet50-smoothing_adversarial_PGD_1step_eps_512_noise_1.00 [72]. Robustness intervention model. `https://github.com/Hadisalman/smoothing-adversarial`

118. resnet50-vtab [112]. Standard training model. `https://tfhub.dev/s?publisher=vtab`

119. resnet50-vtab-exemplar [112]. Standard training model. `https://tfhub.dev/s?publisher=vtab`

120. resnet50-vtab-rotation [112]. Standard training model. `https://tfhub.dev/s?publisher=vtab`

121. resnet50-vtab-semi-exemplar [112]. Standard training model. `https://tfhub.dev/s?publisher =vtab`

122. resnet50-vtab-semi-rotation [112]. Standard training model. `https://tfhub.dev/s?publisher=v tab`

123. resnet50_adv-train-free [74]. Robustness intervention model. `https://github.com/mahyarnajib i/FreeAdversarialTraining`

124. resnet50_augmix [41]. Robustness intervention model. `https://github.com/google-research /augmix`

125. resnet50_aws_baseline. Standard training model.

126. resnet50_cutmix [108]. Robustness intervention model. `https://github.com/clovaai/CutMix-PyTorch`

127. resnet50_cutout [20]. Robustness intervention model. `https://github.com/clovaai/CutMix-PyTorch`

128. resnet50_deepaugment [40]. Robustness intervention model. `https://github.com/hendrycks/imagenet-r`

129. resnet50_deepaugment_augmix [40]. Robustness intervention model. `https://github.com/hendrycks/imagenet-r`

130. resnet50_feature_cutmix [108]. Robustness intervention model. `https://github.com/clovaai/CutMix-PyTorch`

131. resnet50_imagenet_100percent_batch64_original_images. Standard training model.

132. resnet50_imagenet_subsample_125_classes_batch64_original_images. Standard training model.

133. resnet50_imagenet_subsample_1_of_16_batch64_original_images. Standard training model.

134. resnet50_imagenet_subsample_1_of_2_batch64_original_images. Standard training model.

135. resnet50_imagenet_subsample_1_of_32_batch64_original_images. Standard training model.

136. resnet50_imagenet_subsample_1_of_4_batch64_original_images. Standard training model.

137. resnet50_imagenet_subsample_1_of_8_batch64_original_images. Standard training model.

138. resnet50_imagenet_subsample_250_classes_batch64_original_images. Standard training model.

139. resnet50_imagenet_subsample_500_classes_batch64_original_images. Standard training model.

140. resnet50_l2_eps3_robust [27]. Robustness intervention model. `https://github.com/MadryLab/robustness`

141. resnet50_linf_eps4_robust [27]. Robustness intervention model. `https://github.com/MadryLab/robustness`

142. resnet50_linf_eps8_robust [27]. Robustness intervention model. `https://github.com/MadryLab/robustness`

143. resnet50_lpf2 [115]. Robustness intervention model. `https://github.com/adobe/antialiased-cnns`

144. resnet50_lpf3 [115]. Robustness intervention model. `https://github.com/adobe/antialiased-cnns`

145. resnet50_lpf5 [115]. Robustness intervention model. `https://github.com/adobe/antialiased-cnns`

146. resnet50_mixup [113]. Robustness intervention model. `https://github.com/clovaai/CutMix-PyTorch`

147. resnet50_ssl [104]. Trained with more data model. `https://github.com/facebookresearch/semi-supervised-ImageNet1K-models`

148. resnet50_swsl [104]. Trained with more data model. `https://github.com/facebookresearch/semi-supervised-ImageNet1K-models`

149. resnet50_trained_on_SIN [34]. Robustness intervention model. `https://github.com/rgeirhos/texture-vs-shape`

150. resnet50_trained_on_SIN_and_IN [34]. Robustness intervention model. `https://github.com/rgeirhos/texture-vs-shape`

151. resnet50_trained_on_SIN_and_IN_then_finetuned_on_IN [34]. Robustness intervention model. `https://github.com/rgeirhos/texture-vs-shape`

152. resnet50_with_brightness_aws. Robustness intervention model.

153. resnet50_with_contrast_aws. Robustness intervention model.

154. resnet50_with_defocus_blur_aws. Robustness intervention model.

155. resnet50_with_fog_aws. Robustness intervention model.

156. resnet50_with_frost_aws. Robustness intervention model.

157. resnet50_with_gaussian_noise_aws. Robustness intervention model.

158. resnet50_with_gaussian_noise_contrast_motion_blur_jpeg_compression_aws. Robustness intervention model.

159. resnet50_with_greyscale_aws. Robustness intervention model.

160. resnet50_with_jpeg_compression_aws. Robustness intervention model.

161. resnet50_with_motion_blur_aws. Robustness intervention model.

162. resnet50_with_pixelate_aws. Robustness intervention model.

163. resnet50_with_saturate_aws. Robustness intervention model.

164. resnet50_with_spatter_aws. Robustness intervention model.

165. resnet50_with_zoom_blur_aws. Robustness intervention model.

166. resnext101_32x16d_ssl [104]. Trained with more data model. `https://github.com/facebookr esearch/semi-supervised-ImageNet1K-models`

167. resnext101_32x4d [103]. Standard training model. `https://github.com/Cadene/pretrained -models.pytorch`

168. resnext101_32x4d_ssl [104]. Trained with more data model. `https://github.com/facebookres earch/semi-supervised-ImageNet1K-models`

169. resnext101_32x4d_swsl [104]. Trained with more data model. `https://github.com/facebookr esearch/semi-supervised-ImageNet1K-models`

170. resnext101_32x8d [103]. Standard training model. `https://github.com/pytorch/vision/tree /master/torchvision/models`

171. resnext101_32x8d_deepaugment_augmix [40]. Robustness intervention model. `https://github.c om/hendrycks/imagenet-r`

172. resnext101_32x8d_ssl [104]. Trained with more data model. `https://github.com/facebookres earch/semi-supervised-ImageNet1K-models`

173. resnext101_32x8d_swsl [104]. Trained with more data model. `https://github.com/facebookr esearch/semi-supervised-ImageNet1K-models`

174. resnext101_64x4d [103]. Standard training model. `https://github.com/Cadene/pretrained -models.pytorch`

175. resnext50_32x4d [103]. Standard training model. `https://github.com/pytorch/vision/tree /master/torchvision/models`

176. resnext50_32x4d_ssl [104]. Trained with more data model. `https://github.com/facebookres earch/semi-supervised-ImageNet1K-models`

177. resnext50_32x4d_swsl [104]. Trained with more data model. `https://github.com/facebookres earch/semi-supervised-ImageNet1K-models`

178. se_resnet101 [42]. Standard training model. `https://github.com/Cadene/pretrained-model s.pytorch`

179. se_resnet152 [42]. Standard training model. `https://github.com/Cadene/pretrained-model s.pytorch`

180. se_resnet50 [42]. Standard training model. `https://github.com/Cadene/pretrained-models. pytorch`

181. se_resnext101_32x4d [42]. Standard training model. `https://github.com/Cadene/pretrained -models.pytorch`

182. se_resnext50_32x4d [42]. Standard training model. `https://github.com/Cadene/pretrained -models.pytorch`

183. senet154 [42]. Standard training model. `https://github.com/Cadene/pretrained-models.py torch`

184. shufflenet_v2_x0_5 [54]. Standard training model. `https://github.com/pytorch/vision/tree /master/torchvision/models`

185. shufflenet_v2_x1_0 [54]. Standard training model. `https://github.com/pytorch/vision/tree /master/torchvision/models`

186. squeezenet1_0 [45]. Standard training model. `https://github.com/Cadene/pretrained-mod els.pytorch`

187. squeezenet1_1 [45]. Standard training model. `https://github.com/Cadene/pretrained-mod els.pytorch`

188. vgg11 [78]. Standard training model. `https://github.com/Cadene/pretrained-models.pytorch`

189. vgg11_bn [78]. Standard training model. `https://github.com/Cadene/pretrained-models.pytorch`

190. vgg13 [78]. Standard training model. `https://github.com/Cadene/pretrained-models.pytorch`

191. vgg13_bn [78]. Standard training model. `https://github.com/Cadene/pretrained-models.pytorch`

192. vgg16 [78]. Standard training model. `https://github.com/Cadene/pretrained-models.pytorch`

193. vgg16_bn [78]. Standard training model. `https://github.com/Cadene/pretrained-models.pytorch`

194. vgg16_bn_lpf2 [115]. Robustness intervention model. `https://github.com/adobe/antialiased-cnns`

195. vgg16_bn_lpf3 [115]. Robustness intervention model. `https://github.com/adobe/antialiased-cnns`

196. vgg16_bn_lpf5 [115]. Robustness intervention model. `https://github.com/adobe/antialiased-cnns`

197. vgg16_lpf2 [115]. Robustness intervention model. `https://github.com/adobe/antialiased-cnns`

198. vgg16_lpf3 [115]. Robustness intervention model. `https://github.com/adobe/antialiased-cnns`

199. vgg16_lpf5 [115]. Robustness intervention model. `https://github.com/adobe/antialiased-cnns`

200. vgg19 [78]. Standard training model. `https://github.com/Cadene/pretrained-models.pytorch`

201. vgg19_bn [78]. Standard training model. `https://github.com/Cadene/pretrained-models.pytorch`

202. wide_resnet101_2 [109]. Standard training model. `https://github.com/pytorch/vision/tree/master/torchvision/models`

203. wide_resnet50_2 [109]. Standard training model. `https://github.com/pytorch/vision/tree/master/torchvision/models`

204. xception [12]. Standard training model. `https://github.com/Cadene/pretrained-models.pytorch`

# H   Model accuracies

Table 1: Top-1 model accuracies on ImageNet validation set, effective robustness as calculated with respect to ImageNetV2, an average over all the corruptions, and an average over all the pgd attacks. Note that since we take an average of many attacks, the PGD column can no longer be considered a worst-case attacker for the model (look to F.3 for specific attacks).

| Model accuracies | | | | |
|---|---|---|---|---|
| Model | ImageNet accuracy | ImageNetV2 eff. robust. | Avg. corr. accuracy | Avg. PGD accuracy |
| efficientnet-l2-noisystudent | 88.32 | 1.11 | | |
| FixResNeXt101_32x48d_v2 | 86.36 | 0.97 | 65.65 | |
| FixResNeXt101_32x48d | 86.26 | 0.95 | 65.56 | |
| instagram-resnext101_32x48d | 85.44 | 1.26 | 65.53 | 24.1 |
| efficientnet-b8-advprop-autoaug | 85.37 | 0.51 | 71.85 | |
| BiT-M-R152x4-ILSVRC2012 | 85.18 | -0.31 | 67.26 | |
| efficientnet-b7-advprop-autoaug | 85.09 | 0.66 | 68.92 | |
| instagram-resnext101_32x32d | 85.09 | 1.54 | 64.77 | 24.4 |
| BiT-M-R101x3-ILSVRC2012 | 84.78 | -1.35 | 63.44 | |
| efficientnet-b6-advprop-autoaug | 84.76 | 0.75 | 68.65 | 50.67 |
| efficientnet-b7-randaug | 84.73 | 0.11 | 69.12 | |
| efficientnet-b7-autoaug | 84.33 | 0.32 | 62.77 | |
| efficientnet-b5-advprop-autoaug | 84.3 | 0.51 | 67.76 | 50.17 |
| resnext101_32x8d_swsl | 84.29 | 1.19 | 63.17 | 23.22 |
| instagram-resnext101_32x16d | 84.18 | 1.51 | 63.22 | 29.19 |
| BiT-M-R50x3-ILSVRC2012 | 84.15 | -0.76 | 60.23 | |
| efficientnet-b6-autoaug | 84.13 | 0.14 | 63.42 | 34.29 |
| FixPNASNet | 83.7 | -0.0 | 61.35 | 22.8 |
| efficientnet-b5-autoaug | 83.63 | 0.25 | 62.3 | 32.43 |
| efficientnet-b5-randaug | 83.53 | 0.08 | 63.35 | 34.41 |
| resnext101_32x4d_swsl | 83.23 | 1.41 | 60.09 | 21.73 |
| efficientnet-b5 | 83.11 | 0.17 | 60.28 | 35.18 |
| pnasnet5large | 82.74 | 0.21 | 61.76 | 29.46 |
| instagram-resnext101_32x8d | 82.69 | 1.59 | 60.81 | 30.13 |
| efficientnet-b4-advprop-autoaug | 82.69 | 0.42 | 64.88 | 50.72 |
| efficientnet-b4-autoaug | 82.55 | 0.17 | 59.59 | 34.24 |
| BiT-M-R101x1-ILSVRC2012 | 82.52 | -0.42 | 58.28 | |
| nasnetalarge | 82.51 | 0.48 | 61.74 | 36.99 |
| efficientnet-b4 | 82.23 | -0.64 | 57.2 | 37.06 |
| resnext50_32x4d_swsl | 82.18 | 1.26 | 56.38 | 21.09 |
| resnext101_32x16d_ssl | 81.84 | 0.3 | 58.63 | 22.34 |
| resnext101_32x8d_ssl | 81.63 | 0.73 | 57.96 | 20.82 |
| senet154 | 81.3 | -0.07 | 54.11 | 30.65 |
| resnet50_swsl | 81.18 | 1.35 | 53.95 | 21.39 |
| efficientnet-b3-advprop-autoaug | 81.09 | 0.29 | 60.6 | 51.09 |
| efficientnet-b3-autoaug | 81.05 | 0.17 | 55.5 | 31.76 |
| resnext101_32x4d_ssl | 80.93 | 0.48 | 55.65 | 20.54 |
| polynet | 80.86 | 0.36 | 54.02 | 23.05 |
| BiT-M-R50x1-ILSVRC2012 | 80.4 | -0.63 | 52.21 | 12.5 |
| resnext50_32x4d_ssl | 80.33 | 0.44 | 52.57 | 19.75 |
| inceptionresnetv2 | 80.27 | 0.32 | 56.85 | 34.85 |
| se_resnext101_32x4d | 80.24 | 0.47 | 52.26 | 28.77 |
| efficientnet-b3 | 80.21 | -0.48 | 53.31 | 34.22 |
| inceptionv4 | 80.08 | 0.5 | 55.52 | 28.02 |
| resnext101_32x8d_deepaugment_augmix | 79.9 | 0.25 | 65.56 | |
| resnet101_cutmix | 79.83 | -0.39 | 50.15 | 25.6 |
| efficientnet-b2-autoaug | 79.78 | 0.17 | 53.5 | 30.93 |
| Table continues onto next page | | | | |

| Model accuracies (continued from previous page) | | | | |
|---|---|---|---|---|
| Model | ImageNet accuracy | ImageNetV2 eff. robust. | Avg. corr. accuracy | Avg. PGD accuracy |
| FixResNet50CutMix_v2 | 79.76 | -1.21 | 43.44 | 18.19 |
| dpn107 | 79.75 | -0.47 | 52.37 | 30.64 |
| FixResNet50CutMix | 79.74 | -1.22 | 43.39 | 18.14 |
| efficientnet-b2-advprop-autoaug | 79.6 | -0.25 | 55.17 | 46.33 |
| dpn131 | 79.43 | -0.2 | 52.06 | 30.38 |
| dpn92 | 79.4 | -0.65 | 49.29 | 25.69 |
| resnext101_32x8d | 79.31 | -0.34 | 49.68 | 25.38 |
| resnet50_ssl | 79.23 | 0.52 | 50.15 | 20.57 |
| dpn98 | 79.22 | 0.08 | 51.82 | 30.14 |
| google_resnet101_jft-300M | 79.2 | -0.23 | 53.49 | 26.84 |
| FixResNet50_v2 | 79.1 | -0.62 | 43.31 | 15.38 |
| se_resnext50_32x4d | 79.08 | 0.27 | 50.65 | 24.74 |
| FixResNet50 | 79.0 | -0.67 | 43.25 | 15.3 |
| resnext101_64x4d | 78.96 | -0.2 | 52.06 | 23.57 |
| efficientnet-b2 | 78.89 | -0.39 | 50.05 | 33.88 |
| wide_resnet101_2 | 78.85 | -0.87 | 48.2 | 25.24 |
| xception | 78.82 | 0.06 | 51.7 | 26.32 |
| efficientnet-b1-autoaug | 78.72 | -0.07 | 51.19 | 30.69 |
| se_resnet152 | 78.66 | 0.45 | 50.94 | 28.42 |
| resnet50_cutmix | 78.6 | -1.1 | 44.7 | 26.46 |
| efficientnet-b1-advprop-autoaug | 78.54 | -0.23 | 53.7 | 46.54 |
| wide_resnet50_2 | 78.47 | -0.61 | 46.23 | 26.13 |
| se_resnet101 | 78.4 | 0.43 | 50.12 | 28.2 |
| resnet152 | 78.31 | 0.27 | 47.81 | 22.48 |
| resnet101-tencent-ml-images | 78.25 | 0.04 | 47.77 | |
| resnet50_feature_cutmix | 78.21 | -0.42 | 44.33 | 25.36 |
| resnext101_32x4d | 78.19 | -0.13 | 50.96 | 22.38 |
| resnet101_lpf3 | 78.12 | -0.27 | 46.52 | 22.48 |
| efficientnet-b1 | 77.91 | -0.24 | 47.07 | 31.33 |
| resnet101_lpf5 | 77.91 | 0.1 | 46.54 | 23.13 |
| resnet101_lpf2 | 77.8 | 0.3 | 46.06 | 22.01 |
| se_resnet50 | 77.64 | 0.08 | 48.11 | 27.55 |
| resnext50_32x4d | 77.62 | 0.1 | 45.56 | 22.52 |
| resnet50_augmix | 77.54 | -0.53 | 50.78 | 26.01 |
| resnet50_mixup | 77.47 | -0.54 | 48.2 | 21.95 |
| fbresnet152 | 77.39 | 0.02 | 49.98 | 23.4 |
| resnet101 | 77.37 | 0.01 | 46.06 | 21.85 |
| inceptionv3 | 77.32 | 0.29 | 49.83 | 25.72 |
| densenet161 | 77.14 | 0.13 | 49.36 | 22.22 |
| efficientnet-b0-advprop-autoaug | 77.08 | 0.21 | 49.9 | 44.31 |
| resnet50_cutout | 77.07 | -0.65 | 43.81 | 19.8 |
| FixResNet50_no_adaptation | 77.04 | -0.02 | 44.68 | 20.61 |
| dpn68b | 77.03 | -0.28 | 45.67 | 18.7 |
| resnet50_lpf5 | 77.03 | -0.53 | 43.54 | 22.03 |
| densenet201 | 76.9 | -0.12 | 47.63 | 23.95 |
| efficientnet-b0-autoaug | 76.84 | -0.39 | 45.27 | 30.66 |
| resnet50_lpf3 | 76.82 | -0.12 | 43.3 | 21.83 |
| resnet50_lpf2 | 76.79 | -0.25 | 42.22 | 20.91 |
| resnet50_trained_on_SIN_and _IN_then_finetuned_on_IN | 76.72 | -0.04 | 43.96 | 22.78 |
| resnet50_deepaugment | 76.66 | 0.73 | 53.91 | 29.65 |
| efficientnet-b0 | 76.53 | -0.79 | 43.84 | 31.05 |
| resnet50-vtab-rotation | 76.5 | -0.49 | 41.93 | |
| cafferesnet101 | 76.2 | 0.08 | 44.83 | 25.54 |
| resnet152-imagenet11k | 76.18 | 2.09 | 47.33 | 30.64 |
| resnet50_aws_baseline | 76.14 | -0.36 | 42.13 | 21.46 |
| resnet50 | 76.13 | -0.77 | 41.59 | 21.24 |
| Table continues onto next page | | | | |

| Model accuracies (continued from previous page) | | | | |
|---|---|---|---|---|
| Model | ImageNet accuracy | ImageNetV2 eff. robust. | Avg. corr. accuracy | Avg. PGD accuracy |
| resnet50_imagenet_100percent _batch64_original_images | 75.98 | -0.56 | 41.61 | 21.89 |
| dpn68 | 75.87 | -0.56 | 45.46 | 17.71 |
| resnet50_deepaugment_augmix | 75.82 | -0.08 | 58.29 | 33.73 |
| resnet50-randomized_smoothi ng_noise_0.00 | 75.69 | 0.31 | 41.75 | 21.32 |
| densenet169 | 75.6 | 0.19 | 46.67 | 21.79 |
| resnet50-vtab | 75.54 | 0.22 | 43.61 | |
| resnet50_with_brightness_aws | 75.28 | -0.28 | 43.9 | 22.78 |
| resnet50_with_spatter_aws | 75.21 | -0.29 | 42.81 | 22.45 |
| densenet121_lpf3 | 75.14 | -0.35 | 40.48 | 20.01 |
| densenet121_lpf5 | 75.03 | 0.13 | 41.84 | 21.13 |
| densenet121_lpf2 | 75.03 | 0.41 | 41.24 | 20.82 |
| resnet50_with_saturate_aws | 74.89 | -0.27 | 42.4 | 20.46 |
| resnet50_trained_on_SIN_and _IN | 74.59 | 0.55 | 47.91 | 22.96 |
| resnet34_lpf2 | 74.48 | 0.15 | 41.54 | 20.96 |
| densenet121 | 74.43 | 0.13 | 43.54 | 20.01 |
| resnet34_lpf3 | 74.34 | 0.25 | 42.22 | 20.97 |
| vgg19_bn | 74.22 | 0.18 | 37.94 | 16.51 |
| resnet34_lpf5 | 74.19 | 0.46 | 41.22 | 21.09 |
| resnet50-vtab-exemplar | 74.1 | 0.3 | 44.73 | |
| nasnetamobile | 74.08 | -0.29 | 44.78 | 22.89 |
| vgg16_bn_lpf5 | 74.04 | -0.4 | 36.19 | 18.91 |
| vgg16_bn_lpf2 | 74.01 | 0.13 | 36.06 | 17.81 |
| vgg16_bn_lpf3 | 73.92 | 0.5 | 36.33 | 18.33 |
| resnet50_with_frost_aws | 73.78 | 0.29 | 42.39 | 20.96 |
| resnet50_with_jpeg_compressi on_aws | 73.63 | -0.21 | 41.76 | 38.34 |
| bninception | 73.52 | 1.0 | 40.59 | 21.27 |
| mnasnet1_0 | 73.46 | -0.47 | 36.42 | 18.78 |
| vgg16_bn | 73.36 | -0.09 | 35.69 | 16.19 |
| resnet34 | 73.31 | 0.12 | 40.48 | 21.23 |
| resnet18_swsl | 73.29 | 1.74 | 39.95 | 18.79 |
| resnet50_with_gaussian_noise _aws | 72.97 | 0.21 | 45.56 | 43.88 |
| resnet50_with_gaussian_noise _contrast_motion_blur_jpeg_c ompression_aws | 72.72 | 0.05 | 51.8 | 22.91 |
| mobilenet_v2_lpf2 | 72.62 | -0.56 | 34.46 | 17.46 |
| resnet18_ssl | 72.6 | 1.24 | 39.51 | 19.19 |
| mobilenet_v2_lpf3 | 72.57 | -0.23 | 34.78 | 17.6 |
| mobilenet_v2_lpf5 | 72.51 | -0.1 | 34.9 | 17.73 |
| vgg19 | 72.38 | -0.01 | 32.43 | 20.65 |
| vgg16_lpf5 | 72.33 | 0.15 | 31.89 | 19.86 |
| vgg16_lpf3 | 72.19 | -0.19 | 32.18 | 19.37 |
| vgg16_lpf2 | 72.16 | -0.2 | 31.98 | 19.13 |
| resnet50_with_contrast_aws | 72.0 | -0.42 | 40.85 | 17.29 |
| mobilenet_v2 | 71.88 | -0.13 | 33.96 | 17.49 |
| resnet50_with_fog_aws | 71.76 | -0.83 | 37.9 | 17.19 |
| resnet18_lpf3 | 71.68 | -0.43 | 36.84 | 20.17 |
| vgg16 | 71.59 | -0.33 | 31.3 | 20.14 |
| vgg13_bn | 71.59 | 0.01 | 31.76 | 15.16 |
| resnet18_lpf2 | 71.39 | -0.09 | 36.88 | 19.8 |
| resnet18_lpf5 | 71.39 | -0.51 | 36.86 | 20.22 |
| resnet18-rotation-standard_40 | 71.28 | -0.05 | 36.46 | 20.26 |
| vgg11_bn | 70.37 | -0.1 | 31.7 | 18.05 |
| resnet50-randomized_smoothi ng_noise_0.25 | 70.29 | 0.28 | 40.66 | 63.94 |
| vgg13 | 69.93 | -0.27 | 28.53 | 19.32 |
| googlenet/inceptionv1 | 69.78 | 1.01 | 38.84 | 21.85 |
| Table continues onto next page | | | | |

| Model accuracies (continued from previous page) | | | | |
|---|---|---|---|---|
| Model | ImageNet accuracy | ImageNetV2 eff. robust. | Avg. corr. accuracy | Avg. PGD accuracy |
| resnet18 | 69.76 | 0.46 | 35.01 | 19.51 |
| shufflenet_v2_x1_0 | 69.36 | -0.48 | 30.87 | 16.66 |
| resnet18-rotation-worst10_30 | 69.13 | 0.72 | 34.06 | 22.51 |
| vgg11 | 69.02 | -0.33 | 28.61 | 22.38 |
| resnet18-rotation-random_30 | 68.88 | 0.19 | 32.88 | 18.63 |
| resnet18-rotation-worst10_40 | 68.6 | -0.05 | 32.24 | 22.65 |
| resnet50_with_pixelate_aws | 68.5 | 1.17 | 39.58 | 18.85 |
| resnet18-rotation-random_40 | 68.35 | 0.73 | 31.87 | 17.89 |
| facebook_adv_trained_resnext 101_denoiseAll | 68.33 | -0.11 | 40.86 | 41.42 |
| resnet50-smoothing_adversaria l_DNN_2steps_eps_512_noise _0.25 | 67.87 | -0.31 | 40.57 | 62.89 |
| mnasnet0_5 | 67.6 | -0.37 | 27.9 | 17.39 |
| resnet50_with_motion_blur_aws | 67.46 | 1.49 | 38.71 | 15.34 |
| resnet18-rotation-nocrop_40 | 65.37 | 1.23 | 30.1 | 20.5 |
| facebook_adv_trained_resnet1 52_denoise | 65.32 | 0.38 | 37.97 | 39.48 |
| bninception-imagenet21k | 65.24 | 1.78 | 32.8 | 30.3 |
| resnet50-randomized_smoothi ng_noise_0.50 | 64.24 | 0.04 | 39.8 | 61.41 |
| resnet50_with_greyscale_aws | 63.33 | 0.49 | 28.33 | 18.16 |
| resnet50_linf_eps4_robust | 62.42 | 0.53 | 32.37 | 60.3 |
| facebook_adv_trained_resnet1 52_baseline | 62.34 | 0.58 | 35.77 | 37.63 |
| resnet50-smoothing_adversaria l_DNN_2steps_eps_512_noise _0.50 | 62.19 | -0.04 | 39.14 | 59.26 |
| resnet50-vtab-semi-exemplar | 61.62 | 0.98 | 33.85 | |
| resnet50_with_zoom_blur_aws | 61.25 | 1.22 | 33.27 | 13.01 |
| resnet50-vtab-semi-rotation | 60.92 | 0.94 | 26.38 | |
| shufflenet_v2_x0_5 | 60.55 | -0.27 | 23.58 | 16.08 |
| resnet50_adv-train-free | 60.49 | -0.03 | 29.41 | 57.42 |
| resnet50-smoothing_adversari al_PGD_1step_eps_512_noise _0.25 | 60.47 | -0.45 | 37.21 | 58.49 |
| resnet50_trained_on_SIN | 60.18 | 1.4 | 39.42 | 19.25 |
| squeezenet1_1 | 58.18 | 0.12 | 20.18 | 16.08 |
| squeezenet1_0 | 58.09 | -0.26 | 20.17 | 18.06 |
| resnet50_l2_eps3_robust | 57.9 | 0.33 | 31.83 | 56.25 |
| alexnet_lpf2 | 57.23 | -0.38 | 22.54 | 29.09 |
| alexnet_lpf3 | 56.89 | -0.41 | 22.77 | 30.67 |
| alexnet_lpf5 | 56.58 | -0.41 | 22.77 | 31.71 |
| alexnet | 56.52 | -0.28 | 21.55 | 24.08 |
| resnet50-smoothing_adversari al_PGD_1step_eps_512_noise _0.50 | 54.66 | -0.31 | 35.7 | 53.09 |
| resnet50-randomized_smoothi ng_noise_1.00 | 53.12 | 0.12 | 34.93 | 52.11 |
| resnet50-smoothing_adversaria l_DNN_2steps_eps_512_noise _1.00 | 51.87 | 0.23 | 34.43 | 50.95 |
| resnet50_linf_eps8_robust | 47.91 | 1.35 | 23.93 | 46.97 |
| resnet50-smoothing_adversari al_PGD_1step_eps_512_noise _1.00 | 44.28 | 0.2 | 29.84 | 43.57 |
| resnet50_with_defocus_blur_a ws | 31.9 | 1.3 | 18.18 | 9.29 |
| End of table | | | | |

# I  Synthetic robustness correlation with natural robustness

In this section, we investigate which individual synthetic robustness measures are most predictive of natural distribution shift. For each of the synthetic shifts in our testbed, we compute the effective robustness for each model and measure the Pearson correlation coefficients against the effective robustness under each of the natural distribution shifts in our testbed.

Table 2 provides a full list of the correlation numbers, and Figures 17 to 23 show scatter plots of the two highest correlated synthetic shifts for each natural distribution shift. We find that some of the synthetic shifts are more predictive than others, but none have high correlation with all of the natural shifts. For instance, $\ell_p$-robustness has the highest correlation with consistency shifts, but only low correlation with dataset shifts. On the other hand, some image corruptions such as brightness, gaussian blur, defocus blur, and saturate have higher correlation with the dataset shifts. It is worth nothing our testbed indicates that these synthetic measures are not causal, i.e., models trained on brightness, gaussian blur, defocus blur, or saturate do not have significant positive effective robustness on dataset shifts. Further analyzing these fine-grained connections between synthetic and natural forms of distribution shift is an important direction for future work.

Table 2: Pearson correlation coefficients between all synthetic and natural distribution shifts in our testbed. For each distribution shift, effective robustness was calculated using a linear fit on the standard models. The correlation between synthetic and natural effective robustness was then only computed after filtering out the standard models.

| Pearson correlation coefficients | | | | | | | |
|---|---|---|---|---|---|---|---|
| Synthetic shift | ImageNetV2 | ObjectNet | ImageNetVid (pm-0) | YTBB (pm-0) | ImageNetVid (pm-10) | YTBB (pm-10) | ImageNet-A |
| avg_corruptions | 0.25 | 0.06 | 0.6 | 0.5 | 0.65 | 0.52 | 0.02 |
| avg_pgd | -0.04 | -0.19 | 0.3 | 0.35 | 0.84 | 0.7 | -0.12 |
| brightness_in-memory | 0.34 | 0.11 | 0.32 | 0.3 | 0.29 | 0.23 | 0.13 |
| brightness_on-disk | 0.56 | 0.48 | 0.56 | 0.39 | 0.22 | 0.15 | 0.16 |
| contrast_in-memory | 0.15 | 0.07 | 0.14 | 0.04 | -0.61 | -0.5 | 0.14 |
| contrast_on-disk | 0.26 | 0.28 | 0.17 | 0.05 | -0.61 | -0.54 | 0.15 |
| defocus_blur_in-memory | 0.27 | -0.04 | 0.66 | 0.56 | 0.43 | 0.27 | -0.05 |
| defocus_blur_on-disk | 0.39 | 0.39 | 0.65 | 0.49 | 0.28 | 0.17 | 0.12 |
| elastic_transform-memory | 0.14 | -0.12 | 0.49 | 0.42 | 0.75 | 0.63 | -0.15 |
| elastic_transform-disk | 0.3 | 0.21 | 0.57 | 0.41 | 0.65 | 0.58 | 0.01 |
| fog_in-memory | 0.14 | 0.07 | -0.04 | -0.07 | -0.59 | -0.56 | 0.02 |
| fog_on-disk | 0.28 | 0.31 | 0.04 | -0.03 | -0.64 | -0.6 | 0.04 |
| frost_in-memory | 0.15 | -0.12 | 0.42 | 0.44 | 0.54 | 0.42 | -0.02 |
| frost_on-disk | 0.32 | 0.15 | 0.53 | 0.45 | 0.44 | 0.36 | 0.08 |
| gaussian_blur_in-memory | 0.27 | -0.07 | 0.67 | 0.57 | 0.47 | 0.33 | -0.05 |
| gaussian_blur_on-disk | 0.41 | 0.4 | 0.65 | 0.48 | 0.26 | 0.16 | 0.13 |
| gaussian_noise_i-memory | -0.01 | -0.13 | 0.41 | 0.38 | 0.68 | 0.51 | -0.04 |
| gaussian_noise_o-disk | 0.08 | 0.0 | 0.4 | 0.34 | 0.71 | 0.62 | 0.07 |
| glass_blur_on-disk | 0.24 | 0.17 | 0.56 | 0.45 | 0.61 | 0.53 | -0.0 |
| greyscale | 0.3 | 0.17 | 0.11 | 0.29 | 0.09 | -0.06 | 0.04 |
| impulse_noise_i-memory | -0.06 | -0.1 | 0.35 | 0.34 | 0.65 | 0.45 | -0.05 |
| Table continues onto next page | | | | | | | |

| Synthetic shift | ImageNetV2 | ObjectNet | ImageNetVid (pm-0) | YTBB (pm-0) | ImageNetVid (pm-10) | YTBB (pm-10) | ImageNet-A |
|---|---|---|---|---|---|---|---|
| | | Pearson correlation coefficients (continued from previous page) | | | | | |
| impulse_noise_o disk | 0.04 | 0.0 | 0.34 | 0.31 | 0.72 | 0.6 | 0.03 |
| jpeg_compressio memory | 0.04 | -0.11 | 0.43 | 0.41 | 0.8 | 0.62 | -0.01 |
| jpeg_compressio disk | 0.09 | 0.01 | 0.44 | 0.4 | 0.8 | 0.65 | 0.03 |
| motion_blur_in- memory | 0.2 | -0.02 | 0.51 | 0.43 | 0.56 | 0.41 | -0.08 |
| motion_blur_on- disk | 0.32 | 0.25 | 0.58 | 0.43 | 0.39 | 0.31 | 0.07 |
| pgd.l2.eps0.1 | -0.03 | -0.01 | 0.18 | 0.25 | 0.64 | 0.44 | -0.33 |
| pgd.l2.eps0.5 | -0.05 | -0.22 | 0.31 | 0.34 | 0.71 | 0.63 | -0.11 |
| pgd.linf.eps0.5 | -0.05 | -0.23 | 0.28 | 0.33 | 0.84 | 0.7 | -0.13 |
| pgd.linf.eps2 | 0.01 | -0.18 | 0.3 | 0.31 | 0.76 | 0.69 | 0.05 |
| pixelate_in- memory | 0.27 | 0.03 | 0.61 | 0.48 | 0.66 | 0.53 | 0.05 |
| pixelate_on- disk | 0.31 | 0.16 | 0.62 | 0.46 | 0.63 | 0.54 | 0.12 |
| saturate_in- memory | 0.35 | 0.08 | 0.4 | 0.43 | 0.38 | 0.27 | 0.12 |
| saturate_on- disk | 0.55 | 0.43 | 0.46 | 0.41 | 0.26 | 0.16 | 0.13 |
| shot_noise_in- memory | -0.01 | -0.14 | 0.41 | 0.39 | 0.69 | 0.51 | -0.05 |
| shot_noise_on- disk | 0.07 | -0.01 | 0.4 | 0.35 | 0.71 | 0.62 | 0.06 |
| snow_in- memory | 0.26 | 0.02 | 0.39 | 0.35 | 0.61 | 0.55 | 0.04 |
| snow_on-disk | 0.33 | 0.14 | 0.5 | 0.43 | 0.6 | 0.51 | 0.04 |
| spatter_in- memory | 0.09 | -0.06 | 0.36 | 0.36 | 0.8 | 0.66 | -0.08 |
| spatter_on- disk | 0.26 | 0.08 | 0.5 | 0.43 | 0.75 | 0.63 | -0.04 |
| speckle_noise_in memory | 0.0 | -0.13 | 0.43 | 0.39 | 0.71 | 0.55 | -0.04 |
| speckle_noise_o disk | 0.08 | -0.02 | 0.42 | 0.36 | 0.74 | 0.66 | 0.02 |
| stylized_imagene | 0.32 | 0.24 | 0.31 | 0.3 | 0.44 | 0.31 | -0.02 |
| zoom_blur_in- memory | 0.21 | 0.23 | 0.45 | 0.35 | 0.45 | 0.36 | -0.01 |
| zoom_blur_on- disk | 0.26 | 0.21 | 0.55 | 0.41 | 0.49 | 0.39 | 0.0 |
| | | End of table | | | | | |

Figure 17: Plots of the two synthetic distribution shifts with the highest correlation with ImageNetV2, compared similarly to Figure 5.

Figure 18: Plots of the two synthetic distribution shifts with the highest correlation with ObjectNet, compared similarly to Figure 5.

Figure 19: Plots of the two synthetic distribution shifts with the highest correlation with ImageNet-Vid-Robust pm-0, compared similarly to Figure 5.

Figure 20: Plots of the two synthetic distribution shifts with the highest correlation with YTBB-Robust pm-0, compared similarly to Figure 5.

Figure 21: Plots of the two synthetic distribution shifts with the highest correlation with ImageNet-Vid-Robust pm-10, compared similarly to Figure 5.

Figure 22: Plots of the two synthetic distribution shifts with the highest correlation with YTBB-Robust pm-10, compared similarly to Figure 5.

Figure 23: Plots of the two synthetic distribution shifts with the highest correlation with ImageNet-A, compared similarly to Figure 5.

# J   Information on our main figures

## J.1   Constructing beta $\beta$

For each distribution shift, we construct the baseline accuracy function $\beta$ by analyzing the linear relationship between model performance on the original and shifted distributions. In particular, when constructing $\beta$ we only include "standard models," models that had not been designed with any robustness properties in mind or have not been trained on any data other than the standard 1,000-class ImageNet training set. Before constructing the predictor, model accuracies are then transformed according to the logit distribution; this transform assigns greater mass at the tails and experimentally provided the best linear fits. $\beta$ is then simply the linear predictor of the shifted distribution based on the independent variable (the original distribution), computed in this scaled space.

## J.2   Ablations on our main figures

Here we provide various versions of the main figures in the main text. In each plot, we use logit scaling to demonstrate that gains in performance at higher accuracies represent greater progress. The 95% confidence intervals were empirically computed from the bootstrapped samples. The bootstrapping was performed by computing 1,000 linear fits by sampling the models with replacement.

Figure 24: Only standard models are shown in these plots. Otherwise, they are identical to the main plots in the main text. This is done to better illustrate the quality of the linear fit.

Figure 25: The x-axes are not subsampled in these plots (they are performance on the full ImageNet validation set). Otherwise, they are identical to the main plots in the main text. This is done to clarify that subsampling the axes does not skew the discussed results.

Figure 26: The full y=x line is shown here in these plots. Otherwise, they are identical to the main plots in the main text. This is done to illustrate the performance gap due to distribution shift for each of the natural shifts.

# K Example images of distribution shifts in our testbed

## K.1 Natural distribution shift images

Figure 27: Dataset shifts. Examples from ImageNetV2 (first row), ObjectNet (second row), ImageNet-Vid-Robust (third row), and YTBB-Robust (fourth row).

Figure 28: Consistency shifts. Sequences of video frames from ImageNet-Vid-Robust (top) and YTBB-Robust (bottom).

Figure 29: Adversarial shifts. Examples from ImageNet-A.

## K.2 Synthetic distribution shift images

Figure 30: Sample demonstration of the synthetic distribution shifts in our testbed. Note: This list is not complete. See Appendix F for a complete list.

# L  Additional discussion of related work

## L.1  Relationship to other areas in machine learning

**Domain adaptation / transfer learning.**  Our work is focused on generalizing to out-of-distribution data *without* fine-tuning on the target distribution. A complementary approach uses data from the target domain in order to improve generalization on that particular domain [64]. Depending on the scenario, robustness (without fine-tuning) or domain adaptation may be more appropriate. For instance, it may be challenging to record data from the distribution shift, which would prevent fine-tuning before deployment. In some scenarios, we also expect our model to generalize without extra data (e.g., because humans can do so [77]). Concurrent work by Djolonga et al. [21] studies connections between robustness to distribution shifts and transfer learning. Investigating our testbed from the perspective of transfer learning is an interesting direction for future research.

**Domain generalization.**  Out-of-distribution generalization as measured in our robustness testbed is closely related to domain generalization [6, 60]. In domain generalization, the training algorithm has access to samples drawn from multiple different distributions (domains). At test time, the model is evaluated on samples from a new domain that was not present in training. The idea is that having explicit knowledge of multiple domains at training time may help generalization to a new domain at test time.

Several papers have proposed algorithms for domain generalization; we refer to Gulrajani & Lopez-Paz [36] for a comprehensive survey. Our testbed currently does not contain any algorithms explicitly following the domain generalization paradigm (though pre-training on a different distribution and then fine-tuning on ImageNet has similarities to domain generalization). A recent meta-study of domain generalization found that standard empirical risk minimization performs as well or better than the eight domain generalization algorithms they compared to [36]. This result of Gulrajani & Lopez-Paz [36] has similarities to our finding that robustness interventions currently rarely improve over the trend given by standard (ERM) models trained without a robustness intervention. Evaluating domain generalization approaches on the distribution shifts in our testbed may yield new insights into the performance characteristics of these algorithms.

**Distributionally robust optimization.**  Distributionally robust optimization (DRO) is another recently proposed technique to increase robustness to distribution shift [22, 23]. The DRO objective minimizes the worst case risk over all distributions close to the data distribution (or in the group DRO setting, the worst case risk over all defined groups). DRO has been used to train adversarially robust models [79], vision models with higher worst-group accuracies [71], models less reliant on spurious correlations [81], and many others [24, 62]. For a more thorough discussion on DRO and related work, we refer the reader to [22]. We are currently unable to include DRO models as we are not aware of any pre-trained DRO models for ImageNet. We will add DRO models to our testbed as they become available.

**Adversarial filtering.**  One of the distribution shifts in our testbed was obtained via adversarial filtering (ImageNet-A, [39]). Architectures introduced after the model used to filter ImageNet-A made quick progress in closing the accuracy gap (see Section 4.1). A similar phenomenon occurred in natural language processing. Zellers et al. [111] introduced Swag, an adversarially filtered test for grounded commonsense inference, a combination of natural language inference and commonsense reasoning. At the time of publication, the best model achieved 59% accuracy, while a human expert achieved 85%. Two months later, Devlin et al. [19] introduced the BERT model which achieves 86% accuracy on Swag. This provides further evidence that adversarial filtering can create test sets that are only hard for a specific (existing) class of models.

In the context of training sets, adversarial filtering is similar to hard negative mining, which is often used to generate training data for detection models [17, 30, 69, 83]. Bras et al. [8] propose AFLite, an adversarial filtering algorithm for both refining training sets and creating harder test sets. They evaluate AFLite on natural language inference tasks and ImageNet classification. An interesting question is whether combining their algorithm with a ResNet-50 and evaluating later models leads to similar phenomena as on ImageNet-A [39] and Swag [111].

**Fairness in machine learning.**  Mitchell et al. [59] proposed model cards to document the performance of machine learning models in a variety of conditions. Their focus is on human-centered models and distribution shifts arising from demographic groups (race, gender, etc.). Our focus here is on ImageNet due to the large number of available models and distribution shifts, but the underlying

problem is similar: machine learning models are often brittle under distribution shift. We remark that ImageNet is known to have geo-diversity deficiencies [75], among other issues [14, 25]. In the context of OpenImages [51], researchers have proposed the Inclusive Image dataset [1]. Adding OpenImages and Inclusive Images to our testbed and comparing these distribution shifts to our existing examples is an interesting direction for future work.

**Further domains.** Our work is focused on the domain of image classification. There is a long line of work considering robustness (either natural or synthetic) on other domains [26, 52, 58, 80, 106]. In the context of natural language processing, Belinkov & Bisk [4] explore language model robustness to synthetic versus natural one-word substitutions and reach similar high-level results, finding there is limited robustness transfer between the two distributions.

## L.2 ImageNet-R & ImageNet-Sketch

Recently, Hendrycks et al. [40] studied robustness of classifiers to a new dataset that measure distribution shift, ImageNet-R, along with a new data augmentation method, DeepAugment. The authors make a number of comparisons in relation to an earlier version of this manuscript [90]. In order to provide more clarity, we integrate the ImageNet-R dataset and the DeepAugment models into our testbed in this paper.

Figure 31: Model accuracies on two datasets: ImageNet-R (left), and ImageNet-Sketch (right). Both datasets create a distribution shift by selectively sampling images of renditions or sketches, respectively. Evaluations on these distribution shifts are similar to each other and follow the high-level trends of the other natural dataset distribution shifts in our testbed, with models trained on extra data providing the most robustness (though the effect is not uniform). On the left plot, DeepAugment models are highlighted in dark brown squares, and ImageNet classes were subsampled to match the class distribution of ImageNet-R; the ImageNet-Sketch class distribution already matches ImageNet. Confidence intervals, axis scaling, and the linear fit are computed similarly to Figure 2.

**ImageNet-R.** In Figure 31, we plot model accuracies on ImageNet-R(endition) [40] and a similar dataset of sketches, ImageNet-Sketch [95]. We find that a few of the models trained on more data substantially outperform the rest. The top-right green cluster on both plots consists of several ResNet and ResNeXt models trained on 1 billion Instagram images [56, 92, 104] and EfficientNet-L2 (NoisyStudent) trained on the JFT-300M dataset of 300 million images [102].

However, as with the other dataset shifts, not all models trained on more data follow this trend. Several ResNet models trained on either the YFCC 100 Million images dataset [104] or the full ImageNet 21k-class dataset [49] have close to zero or negative effective robustness.

When interpreting the results of models trained on more data, a caveat is that the extra training data may contain renditions that do not occur in ImageNet. To clarify this point, we have reached out to the authors of [56] to obtain more information about the Instagram dataset. We will update our paper when sufficient data becomes available to estimate the relative frequency of renditions in the Instagram dataset. In the meantime, we note that the performance of the Instagram-trained models gives an answer to a question between the following two extremes: (i) How much performance on ImageNet-R do current models gain from a large, uncurated set of social media images that contains

Figure 32: We compare the effective robustness of models with their accuracy drop due to corruptions (left) and adversarial attacks (right). The effective robustness is computed with respect to the linear fit on ImageNet-R. The measures are weakly correlated, indicating that improved robustness to corruptions or adversarial attacks does not in general improve effective robustness under ImageNet-R.

renditions? (ii) How much robustness to ImageNet-R do current models gain from a large, uncurated set of social media images that contains little to no renditions?

Interestingly, we also find that a number of $\ell_p$-adversarially robust models provide substantial effective and relative robustness on both datasets. The top-left cluster of three yellow points on both plots are feature denoising models trained by Xie et al. [101]. These results suggest that adversarial robustness and denoising blocks can be viable approaches for distribution shift comprised of renditions or sketches.

A natural follow-up question is whether synthetic robustness is correlated with ImageNet-R robustness. Similar to Appendix D.2, in Figure 32 we compare effective robustness on synthetic distribution shifts against effective robustness on ImageNet-R. The scatter plots are weakly correlated (the Pearson correlation coefficients are $r = 0.35, 0.30$), indicating that improved robustness to corruptions or adversarial attacks in general does not improve effective robustness on ImageNet-R. However, there does appear to be a strong trend for the brown points in the left plot. Indeed, the correlation coefficient computed for only the "other robustness intervention" models is $r = 0.76$, suggesting that for this category of models, image corruptions robustness is well correlated with ImageNet-R robustness.

**DeepAugment.** Hendrycks et al. [40] also introduce a new data augmentation method, DeepAugment, which generates data augmentations by distorting the weights and activations of an image-to-image translation network. As seen in Figure 31, the DeepAugment+Augmix models, the top two dark brown squares on the left plot, have higher effective robustness on ImageNet-R than most other models ($\rho = 11.2\%$ for ResNeXt101 and $\rho = 10.2\%$ for ResNet50).

As mentioned above, some of the models trained on the large Instagram and JFT-300M datasets outperform all other approaches on ImageNet-R including DeepAugment, but it is unclear how many images of renditions these datasets contain. Among the other models trained only on ImageNet, the comparison between $\ell_p$-adversarial robustness and DeepAugment is nuanced. The $\ell_p$-robust model of [101] has higher effective robustness but reduces standard ImageNet accuracy. The highest accuracy on ImageNet-R is also achieved by a model with an $\ell_p$-based robustness intervention (an AdvProp model [100]), but the model is derived from EfficientNet [88] which achieves higher standard accuracy than the wide ResNeXt model [103] used in DeepAugment. An interesting question for future work is how and why $\ell_p$-robustness helps on ImageNet-R, e.g., by training a ResNeXt model with AdvProp.

On the anchor frames of ImageNet-Vid-Robust and YTBB-Robust, DeepAgument provides effective robustness comparable to models trained on multiple synthetic perturbations (e.g., a combination of Gaussian noise, contrast, motion blur, and JPEG compression). On ImageNetV2 and ObjectNet, DeepAugment does not provide effective robustness.

For ImageNet-C corruptions [38], the combination of DeepAugment and AugMix offers substantial robustness. Excluding $\ell_p$-adversarially robust models in the low accuracy regime, the two models

with highest effective robustness to ImageNet-C corruptions are DeepAugment- and Augmix- trained ResNet50 ($\rho = 14.2\%$) and ResNeXt101 ($\rho = 13.2\%$). To put this in context, a ResNet50 trained directly on four of the ImageNet-C corruptions (Gaussian noise, contrast, motion blur, and JPEG compression) achieves an effective robustness of $\rho = 13.3\%$.

## L.3 Video robustness

In the context of video robustness, Gu et al. [35] have previously measured the performance of image classifiers on video sequences from the YouTube-BoundingBoxes(YT-BB) dataset [67]. They find that video robustness is strongly correlated with accuracy on color corruptions such as brightness, hue, and saturation, with correlation coefficients $r$ near $0.95$. There are two potential reasons our testbed finds these measures to be only weakly correlated with robustness on YTBB-Robust and ImageNet-Vid-Robust ($r$ ranging from $0.1$ to $0.5$ - full table in Appendix I):

**Standard accuracy as a confounder.** The analysis in [35] does not account for standard accuracy as a confounder. The authors consider video robustness as accuracy within k frames of an anchor frame given that the anchor frame was correctly classified. While this definition does somewhat account for models with higher standard accuracies, it is natural to expect that models with higher standard accuracy are still more likely to predict the neighboring frames correctly given that the anchor frame was predicted correctly. Thus, standard accuracy will be correlated with video robustness. Moreover, our testbed reveals that standard accuracy is also correlated with corruption accuracy, and hence corruption accuracy will be correlated with video robustness as well.

Additionally, it is worth noting that correlation does not mean that robustness to color corruptions *cause* robustness on videos. For instance, our testbed contains a model trained on saturation as data augmentation. While this model is highly robust to saturation (exhibiting only a 1% drop from standard accuracy to accuracy under saturations, compared to a baseline model exhibiting a 12% drop), it is no more video robust than a baseline without the saturation training (the saturation-trained model still experiences an 18% video robustness drop, compared to a baseline model exhibiting a 19% drop). This example further shows the need for our measure of effective robustness as it explicitly corrects for the confounding effect of standard accuracy.

**Differences in data preparation.** Gu et al. [35] split the full YT-BB dataset into training, validation, and testing splits, and evaluate ImageNet models on sequences from the test split. In contrast, Shankar et al. [76] derive datasets from ImageNet-Vid and Youtube-BB through a rigorous cleaning process: inspecting and annotating each sequence with human experts to check that subjects appear in frame throughout the sequence, match the correct class, and are not very blurry. This cleaning process indicates the derived datasets (ImageNet-Vid-Robust and YTBB-Robust) are better calibrated to measuring classification performance.

## Footnotes

[3]Each ResNet50 was trained with a batch size of 256 for 120 epochs, starting with a learning rate of 0.1 and decaying by a factor of 10 every 30 epochs. For the ResNet50s trained on corruptions, we randomly sample a corruption and severity for each image. Refer to F.2 for details on corruptions and severities. We use our custom fast gpu implementations of these corruptions for training.