[Reviews · NeurIPS 2020]

Review 1

Summary and Contributions: The paper complements the synthetic robustness tests with experiments on distribution shifts arising from real data.

Strengths: The key strength of the paper is in the questions it raises - whether robustness measures undertaken on synthetic perturbation datasets can translate into robustness when subjected to natural shift in data distribution.

Weaknesses: The study focuses primarily on evaluating whether or not robustness guarantees can scale from artificially perturbed modifications into more naturally obtained shifts. As such, the study relies on experiements on ImageNet for most part. This is inadequate in my opinion since ImageNet itself is a fairly well-curated dataset and a study of how perturbation based training allows adaptation to natural variations is not very well-placed in a dataset that effectively restricts the extent of such natural variations. It would help to use a slightly more diverse range of datasets to better put the claims in context. This is of significant consequence in several real-world applications of ML algorithms, such as fairness in face recognition etc, and as such ideas about transferring robustness in vision needs to be studied on a broad range of datasets. This is vital in today's era of extensive scrutiny on several aspects of machine learning research and application (this is recognised by the authors indirectly in their Broader Impact section, to their credit) The extent of shifts in distribution considered, in terms of consistency, dataset and adversarial filtering. A range of shifts for the synthetic distributions is considered as well. My question is why can't the shift in distribution be addressed by methods such as transfer learning or knowledge distillation or similar approaches? While robustness evaluation is in itself a challenging topic given the extent of possible synthetic modifications that may emerge for datasets, the quantification of whether or not there is a transfer to natural shifts would still be insufficient.

Correctness: Empirical studies seem to be in the right track, but there is a requirement for further experimentation to support the extent of the claims proposed with regards to the extent of transfer of robustness. Specifically, the studies need to expands beyond ImageNet.

Clarity: The paper is generally readable and explains the claims and experiments in a fair amount of detail

Relation to Prior Work: Yes - fairly well discussed.

Reproducibility: Yes

Additional Feedback: Please consider expanding beyond ImageNet. The conclusions of the study will then be better placed for further research by the community. --- AFTER REBUTTAL---- I have amended the score, after reading other reviews and a particularly strong advocacy by one reviewer.


Review 2

Summary and Contributions: This paper studies the robustness of image classifiers to shifts in the input distribution. The authors propose two new metrics, effective robustness and relative robustness. They conduct a large empirical analysis and find that many robustness interventions do not improve robustness when controlling for accuracy on the original test set, save for training on additional, diverse data. They also find that improvements in effective robustness on synthetic and natural distribution shifts do not correlate.

Strengths: The empirical evaluation is very extensive. It includes hundreds of models and numerous robustness benchmarks, allowing the authors to draw connections at a higher level than previous works, such as the finding that effective robustness on synthetic and natural distribution shifts do not correlate. This particular observation is new and valuable to the research community. To my knowledge, the proposed metrics of effective and relative robustness are the first proposed means of controlling for clean accuracy in the evaluation scenario where we do not have access to clean and corrupted versions of each image. This evaluation scenario will become more commonplace as research moves away from synthetic corruptions, so the metrics are good contributions.

Weaknesses: The paper does not propose any promising methods to improve effective robustness beyond a general recommendation to train on additional, diverse data. This point is explicitly addressed in the paper; the authors note that their focus is not to introduce a new method, but rather to "identify overarching trends that span multiple evaluation settings" (lines 60-61). However, the trend that training on more data helps is a somewhat shallow finding. The paper would be stronger with more analysis of the kinds of methods and architectures that improve robustness. Out of the hundreds of models evaluated, there are surely some deeper insights to be found...? There is no discussion of domain adaptation or domain generalization. Given this paper's focus on natural distribution shifts, this seems like an oversight. In particular, a common finding in domain generalization papers is that the number of source domains is a major factor in accuracy on the target domain. This is very similar to the finding of this paper that "Training on more diverse data improves robustness" (line 52). Some aspects of the proposed testbed are worrisome. The authors propose to measure effective robustness with a log-linear predictor trained on the accuracies of hundreds of models. This seems rather cumbersome, although I understand the provided codebase is designed to be easy to use. The paper does a good job of explaining why robustness measures should control for accuracy, but it doesn't inspire confidence that the proposed way of doing so is optimal. Additionally, the log-linear model is substituted for a piece-wise model for ImageNet-A experiments. The authors explain how this is due to the special nature of ImageNet-A, but what if the log-linear model doesn't fit a different dataset (e.g. one proposed in the future)? Should the person using the proposed testbed use a piece-wise model? Based on the paper, it isn't clear.

Correctness: The new metrics seem appropriately designed. The methodology is sound.

Clarity: The paper is well-written.

Relation to Prior Work: For the most part, this paper situates itself well in relation to prior work. The main difference, which is clearly highlighted throughout the paper, is simultaneously evaluating many models (more than before) on synthetic and natural distribution shifts (not done before). There is no discussion of how the proposed metrics relate to the mCE and relative mCE metrics proposed in [24]. See the point in the "weaknesses" section about domain generalization.

Reproducibility: Yes

Additional Feedback: The included codebase seems easy to use and extend. This is important given the complexity of the proposed testbed. Regarding the proposed relative robustness metric, the correct baseline to use for a given model is not specified. In the appendix, a standard ResNet is used for a swath of experiments, but no other mention is made of how to choose a baseline model. How should one proceed if, for example, the robustness intervention is a new architecture? Regarding the finding that effective robustness on natural and synthetic distribution shifts do not correlate, I would like to see a more granular analysis. Are there certain synthetic shifts for which effective robustness does correlate with effective robustness on natural shifts? Also, some of the correlations in section D.2 are fairly strong (e.g. ImageNet-Vid-Robust pm-0). Do you have any hypothesis for why this is the case? _______________________________ Update after author feedback: The authors and R4 did not address my concern about the dearth of discussion on domain generalization. Note that this is not the same as domain adaptation / transfer learning. The difference is that one does not have access to examples from the target domain, although one may have access to multiple source domains. There are a number of papers that look into this problem setting, and one of the consistent findings in these papers is that training on more, diverse data is a dominant factor in performance (similar to one of the main findings of this paper). While I stand by my claim that surely there are some deeper insights to be found about what kind of interventions help (i.e. a more granular taxonomy than 'robustness method' vs 'more data'), this may be paramount to asking the authors to develop an entirely new method, which would be unreasonable. R4 makes good points regarding the relative worth of this paper. I hope the authors take my remaining concerns into account, but I now believe the paper deserves a 7.


Review 3

Summary and Contributions: The paper presents an in-depth study of robustness to “distribution shifts arising from natural variations in the dataset”. It does so by looking at the performance of a large number of models across an equally large number of benchmark datasets. These datasets cover four “modes” of robustness. Robustness to distribution shifts in images from ImageNet-like distributions (ImgeNet-V2, ObjectNet, ImageNet-vid-robust and YTBB-robust), robustness to small variations in video (pm-k metric on ImageNet-vid-robust and YTBB-robust), adversarially filtered natural images (ImageNet-A) and images with synthetic corruptions (ImageNet-C). Similarly the tested models cover different approaches including standard models, models trained with more data, adversarially trained models and models with robustness interventions. Key findings include the strong dependence of robustness on “natural” distribution shifts to model accuracy (with a knee-like relationship for adversarially filtered images), the absence of any meaningful correlation between “synthetic” and “natural” distribution shifts, the positive effect of lp-adversarial training on the effective robustness towards consistency shifts and the small but noticeable positive effect of training on more diverse data as well as a large number of smaller and more nuanced findings.

Strengths: It is impossible to mention all the strengths of the paper. The systematic study design and the amount of data alone make this a landmark study for robustness research. To still try to capture what I like about the paper I’ll try a quick rundown: - The topic of robustness is timely and definitely very relevant for machine learning as a whole field. It is probably the single most important topic researchers currently have to answer before ML models can be widely deployed. - The method – conducting a large-scale meta-study – is definitely appropriate and sheds light on a range of open questions. The selected models and distribution shifts cover (almost) anything one could wish for (I miss BiT-JFT but that model is an internal Google model and not publicly available). - The findings are relevant and contribute to a growing body of concurrent works that show very similar effects. Among all those studies this is the most extensive and thus potentially the most relevant one. ### Post-rebuttal comments: After reading the other reviews I feel that I have to note the insights this paper created more clearly. I think the paper provides three key insights all of which are important and new: 1. Clean accuracy is by far the best indicator for performance on datasets with "natural" distribution shifts 2. Robustness to adversarials, corruptions and "natural" distribution shifts are not the same 3. 9 years after an "Unbiased Look at Dataset Bias" has appeared we have still not understood much more about distribution shifts All three are all in a way negative results and thus not very surprising but nevertheless important and probably an even bigger contribution than the typical "positive" result as they are harder to obtain (It requires testing many models to make each point convincingly).

Weaknesses: The paper has a number of issues which are beyond the authors control. Most are caused by the extreme publication speed in the field and the limitations of the conference publishing model. Indeed all of my major concerns are of this type: 1. The use of ImageNet-V2 as primary example: The main issue I find in the paper is the use of ImageNet-V2 as the primary example for “natural distribution shift”. There is indeed a large gap between model performance on ImageNet and ImageNet-V2 but almost all of this can be attributed to subtle effects that arise in the dataset replication process as described by Engstrom et. al. 2020. Using this to criticize the paper is however unfair as Engstrom et. al. 2020 was published on 19. May 2020 only a week before the abstract deadline. ### Post-rebuttal comments: I did not know Shankar et. al., ICML ’20. Thanks for pointing out that reference. I have to look more deeply into this but judging from a quick read their results do indeed change my perception on the performance gap in ImageNet-V2. Nevertheless I think ObjectNet is the more obvious example and should be front and center. As Djolonga et. al. 2020 show it's the least correlated with other benchmarks with a very different design making it (in my current view) the most interesting of the selected benchmarks. 2. The definition of “distribution shifts arising in real data”: While the distribution shift from ImageNet to ImageNet-V2 has mostly been explained by Engstrom et. al. 2020 those to ObjetNet, ImageNet-vid-robust and YTBB-vid-robust can reasonably be expected to be real and existent. They do however only cover a subset of distribution shifts arising in real-world images. As to what is missing take for example ImageNet-R which was recently introduced by Hendrycks et. al. 2020. ImageNet-R shows real-world images not of the original objects but of different artistic renditions like paintings or sculptures. In this case some robustness interventions do have an effect. So the discussion of what constitutes a “natural distribution shift” has to be more nuanced. But as before this information was not available to the authors at time of submission because Hendrycks et. a. 2020 was published 3 weeks after the submission deadline. What turns this situation absurd is the fact that Hendrycks et. al. 2020 heavily builds upon what can readily be assumed to be an earlier version of the present article (not cited here to keep the double-blind mechanism as much intact as possible). ### Post-rebuttal comments: Thanks for including ImageNet-R even though it doesn't make the story easier. The dedication to completing the testbed is really amazing. 3. Big Transfer (BiT) models are missing from the analysis: The recently published Big Transfer model family (Kolesnikov et. al. 2019) was shown to have stunning generalization properties. The most interesting model of that family, BiT trained on the JFT300 dataset, has however not yet been publicly released. After seeing the L2-NoisyStudent model perform so well it would have been interesting to see if BiT-JFT can live up to it’s hype. Instead the authors of BiT have released their own robustness study using partly similar methodology as in the paper presented here (Djolonga et. al. 2020). This should not imply that Djolonga et. al. 2020 is biased or does something wrong but simply illustrate how fast paced the field has become. ### Post-rebuttal comments: Thanks for including BiT-M and reaching out to the authors. As I said above the commitment to completeness is great! 4. Too much information for 8 pages: It is pretty obvious that the amount of content presented in this paper is more than fits 8 pages in the NeurIPS template. I think the authors did a good job presenting their work in that format but when reading the paper it is still noticeable that there was much more content than could fit. It becomes even more obvious when reading the appendix which is full of exciting experiments that provide valuable information but have a good chance of being overlooked there. I want to repeat here that these problems are beyond the authors control. Most of it is caused by the huge amount of related work that was done in parallel and the conference submission system makes it impossible to publish longer papers or significantly update them during the review process. As a result I think it would be unfair to judge the submission based on these flaws. I would still appreciated if the authors could adapt their interpretations and related work prior to submission. Thus the following suggestions contain points regarding the above mentioned issues: 1. Use ObjectNet instead of ImageNet-V2 as the go to example (especially in Figures 1 & 5 as well as in Section 4) 2. Discuss different possibilities to select distribution shifts in real data in the introduction. State and motivate your choice. If I was asked I’d call them distribution shifts to ImageNet like images, as opposed to sketches, renditions, images with specific environmental factors like nighttime scenes or images taken in bad weather etc. which have a specific and easy to point out distribution shift. I think the second to last paragraph of the broader impact statement does a good job in justifying and contextualizing this approach and could be used here. 3. Follows thereof: Try to be a bit more specific as the chosen “natural” distribution shifts are just a subset of what is possible. The paper sometimes reads as if it covered all natural distribution shifts while it doesn’t (especially in section 1). Smaller suggestions: 4. Place “Dataset shifts” before “Consistency shifts” in section 3.1.1 or mention the video datasets in the first section 5. Change the description of “Image corruptions” in 3.1.2 removing the statement that you used corruptions from Geirhos et. al. 2019 which according to the appendix were not used and either don’t mention the number of corruptions (38) or explain why it’s 38 and not 19. I’d probably just remove that number as the nuanced discussion of “in memory” and “on disk” corruptions is only mentioned in the appendix. 6. Specify which dataset \rho is computed on in section 41. “Dataset shifts”

Correctness: From what I can judge all methods and claims appear to be correct. Additionally the authors submitted the code for the model testbed which would in theory allow reproducing the results.

Clarity: The paper is mostly well written. Problems however arise from the extreme information density. See corresponding points and suggestions in the “weaknesses” section above.

Relation to Prior Work: Prior work is well discussed and extensively cited. The list of links for every single tested model deserves a special mention (Appendix Section G).

Reproducibility: Yes

Additional Feedback: I found it incredibly difficult to judge this paper. On the one hand this is an excellent and extremely deep study of one of the most pressing issues in machine learning right now. On the other hand the number of weaknesses is quite large but almost none of them were in the authors control. Were this a traditional journal review process I’d vote for a major revision and ask the authors to incorporate the very relevant recent results to ultimately arrive at a publication which is as correct and complete as possible. Importantly this would not require any additional experiments but could be done with some rewriting and in part an extension of the paper in length (there is definitely enough content to fill more pages). As this is not a journal review process but a conference review with almost no room for additional changes I decided to give the publication a very high score despite the weaknesses. I think this is fair for multiple reasons: 1. This is a very interesting and extensive study 2. Most of the weaknesses were caused by closely related work being published in parallel in a very short window of time 3. A major reason all these pieces of related work could appear alongside this paper can be traced to the fact that what I have to assume was an earlier version of this work was rejected from ICLR for reasons I can’t understand (the open review process makes it possible to see the reviewers comments). Had this not happened the paper would have come half a year before the related work and especially before the distribution shift in ImageNet-V2 was explained. At that time the amount of content wouldn’t have been so much too much for 8 pages (even though it was already a lot to unpack back then). I still hope the authors incorporate the newly available information into their writeup and publish an extended version either as preprint or in an appropriate journal. But when I have to decide between seeing this paper being rejected again because of flaws beyond the authors control or being published despite these flaws I’d rather see it published. References: Engstrom et. al. 2020: Identifying Statistical Bias in Dataset Replication, ArXiv 2020 Hendrycks et. al. 2020: The Many Faces of Robustness: A Critical Analysis ofOut-of-Distribution Generalization, ArXiv 2020 Kolesnikov et. al. 2019: Big Transfer (BiT): General Visual Representation Learning, ArXiv 2019 Djolonga et. al. 2020: On Robustness and Transferability of Convolutional Neural Networks, ArXiv 2020 Geirhos et. al. 2019: ImageNet-trained CNNs are biased towards texture; increasing shape bias improves accuracy and robustness, ICLR 2019 ### Post-rebuttal comments: My judgement did not change after seeing the other (much more negative) reviews and reading the rebuttal. One of the main concerns of the other reviewers seem to be that methods like transfer learning or domain adaptation are missing. While these methods are great and could solve the problem on these benchmarks but for real-world applications we want to have models that can generalize without these additional steps. Therefore these methods are not widely considered as the solution to robustness issues despite their efficacy. The other two main concerns are the consideration of only ImageNet style benchmarks and that it is not obvious that a log-linear fit is optimal. While I disagree with the first point because this is the standard approach in the filed at the moment, I think the second point is valid. But that does not mean the method is wrong. It looks like a log-linear fit is a great predictor for all tested benchmarks excluding ImageNet-A. There the identification of the picewise-log-linear relationship is a key insight into the effect of adversarial filtering. And for future works and benchmarks I think it is ok to expect that the authors of a hypotetical future paper think about their results before simply using a log-linear fit as proposed here. In contrast one could even say that showing a log-linear fit is a bad predictor may be a good way to identify an interesting new benchmark. Taking these points into account I still consider this to be an extremely important publication and strongly argue for publication. I also want to positively note the authors willingness to complete their testbed by including another benchmark (ImageNet-R) that came out only after the submission deadline!

[Author Response · NeurIPS 2020]

We thank the reviewers for their feedback and reply to the major points raised by each reviewer individually.

**Reviewer 1**

Our paper focuses on ImageNet classification because this is what almost all prior work on robustness has studied. While
we agree that considering robustness across a wider range of tasks would be an improvement, most prior publications
have focused on ImageNet exclusively as well. ImageNet thus provides a standard benchmark, which is important since
our paper is a meta-study of prior work. Further, since many of the papers that study only ImageNet were publishable
results, we believe this indicates our ImageNet analysis is also sufficient.

We also agree that evaluating on uncurated datasets is an extremely interesting area for future work, especially for the
real-world deployment of machine learning. However, even in the simpler setting when both training and test sets are
well curated, we find that most techniques still do not improve robustness beyond standard accuracy.

We agree that transfer learning is an interesting setting for future work, but note transfer learning cannot always be the
solution, e.g., when the shift can not be reliably quantified or it is hard to collect data from the shifted distribution. To
reiterate our core argument: studying the robustness of models to small changes in distribution is the main focus of our
work. We hope that future work (e.g., transfer learning research) can build on our testbed.

**Reviewer 3**

Our results are substantially more nuanced than "more data helps": (i) We show that *only* more data currently helps
robustness on ImageNetV2 and ObjectNet. Many robustness interventions have been proposed over the last few years,
but they do not help on these two distribution shifts. This is a strong negative result. (ii) Appendix B demonstrates that
additional data *from the same distribution* does *not* help robustness. (iii) The effect of off-distribution data varies. For
instance, one model trained on JFT-300 [57] does not have effective robustness on ImageNetV2 or ObjectNet, while
another model trained on JFT-300 [74] does. Beyond the effect of data, we highlight that $\ell_p$-robust models provide
effective robustness on consistency shifts. Appendix D contains additional results for more granular trends.

Regarding domain adaptation / transfer learning, please refer to the third paragraph in our response to Reviewer 1.

We agree with the concern about the choice of baseline function and will clarify this point. The baseline function
depends on the trend given by the models without robustness intervention; any regression method that fits standard
models well can be used. In practice for the datasets we considered, we found the (piecewise) log-linear fit to be best.

**Reviewer 4**

We greatly appreciate R4's detailed feedback. We concur that the pace of work in this area is a key challenge: our work
is the result of multiple iterations of peer review, each which led to additional models, datasets, and analyses in the
manuscript. One side effect, as R4 points out, is that the manuscript is dense, despite our best attempts at prioritizing
the key analyses for the main paper and leaving details to the supplementary material.

**ImageNet-v2 as primary example.** Another concurrent work, Shankar et. al., ICML '20, shows that humans can
classify ImageNet-v2 images as well as ImageNet: that is, humans do not suffer from a drop in accuracy due to the
distribution shift in ImageNet-v2, while current vision models do. Thus, even if Engstrom et. al.'s analysis explains the
accuracy drop due to selection frequency bias, we would like models that match the robustness of human labelers.

**The definition of "distribution shifts arising in real data":** We included a diverse array of natural distribution shifts
available at the time of submission. As R4 notes, ImageNet-R was released after submission and is a welcome response
to our call for more evaluations on natural distribution shifts. We have added ImageNet-R to our testbed (see Fig. 1) and
will include it in the camera-ready. We would also be happy to include any other natural shifts that we may have missed.

**BiT-L models:** As the BiT-L model is private, we have reached out to the authors to include it in our test bed, and are
simultaneously working on incorporating the smaller, publicly released BiT-M model.

Figure 1: Model accuracies on ImageNet-R, a dataset with renditions such as art, sketches, and graphics of 200 ImageNet classes. Evaluations on this distribution shift follow the high-level trends of the other natural distribution dataset shifts in our testbed, with models trained on extra data providing the most robustness (though the effect is not uniform). ImageNet classes were subsampled to match the class distribution of ImageNet-R. Confidence intervals, linear fit, and axis scaling are computed similarly to Figure 2 in the main text. The color coding is identical to Figure 3 of the main text.

[Meta-Review · NeurIPS 2020]

This paper is a meta-analysis that explores the combinations (1) network architectures, (2) various (adversarial) robust training methods and (3) variants of ImageNet dataset with the goal of quantifying how robust these networks are to various perturbations. All reviewers appreciated the comprehensiveness of the study and belielve that this paper will serve as a starting point / benchmark of studying robustness of image classification models.